# A multicenter clinical AI system study for detection and diagnosis of focal liver lesions

Hanning Ying[1,29], Xiaoqing Liu [ID][2,29], Min Zhang[3,29], Yiyue Ren[4,29], Shihui Zhen[4,29], Xiaojie Wang[4,29], Bo Liu[2,29], Peng Hu[5], Lian Duan[1], Mingzhi Cai[6], Ming Jiang[7], Xiangdong Cheng [ID][8], Xiangyang Gong [ID][9], Haitao Jiang[8], Jianshuai Jiang[10], Jianjun Zheng[11], Kelei Zhu[12], Wei Zhou[13], Baochun Lu[14], Hongkun Zhou[15], Yiyu Shen[16], Jinlin Du[17], Mingliang Ying[17], Qiang Hong[18], Jingang Mo[19], Jianfeng Li[20], Guanxiong Ye[21], Shizheng Zhang[5], Hongjie Hu[5], Jihong Sun[5], Hui Liu[22], Yiming Li[2], Xingxin Xu[2], Huiping Bai [ID][2], Shuxin Wang[2], Xin Cheng[23], Xiaoyin Xu[24] ✉, Long Jiao[25] ✉, Risheng Yu[26] ✉, Wan Yee Lau[27] ✉, Yizhou Yu [ID][28] ✉ & Xiujun Cai [ID][1] ✉

Early and accurate diagnosis of focal liver lesions is crucial for effective treatment and prognosis. We developed and validated a fully automated diagnostic system named Liver Artificial Intelligence Diagnosis System (LiAIDS) based on a diverse sample of 12,610 patients from 18 hospitals, both retrospectively and prospectively. In this study, LiAIDS achieved an F1-score of 0.940 for benign and 0.692 for malignant lesions, outperforming junior radiologists (benign: 0.830-0.890, malignant: 0.230-0.360) and being on par with senior radiologists (benign: 0.920-0.950, malignant: 0.550-0.650). Furthermore, with the assistance of LiAIDS, the diagnostic accuracy of all radiologists improved. For benign and malignant lesions, junior radiologists' F1-scores improved to 0.936-0.946 and 0.667-0.680 respectively, while seniors improved to 0.950-0.961 and 0.679-0.753. Additionally, in a triage study of 13,192 consecutive patients, LiAIDS automatically classified 76.46% of patients as low risk with a high NPV of 99.0%. The evidence suggests that LiAIDS can serve as a routine diagnostic tool and enhance the diagnostic capabilities of radiologists for liver lesions.

Liver cancer is one of the most important liver diseases and the second leading cause of cancer-related deaths worldwide[1]. Hepatocellular carcinoma (HCC) is a dominant liver cancer accounting for 90% of primary liver cancers, and its global incidence continues to rise in comparison to most other cancers[2]. In addition to HCC, common focal liver lesions (FLLs) also include malignant lesions such as intrahepatic cholangiocarcinoma (ICC) and hepatic metastasis (HM), as well as benign lesions such as hepatic cyst (HC), hepatic haemangioma (HH), focal nodular hyperplasia (FNH), and hepatic abscess (HA). Early detection and accurate diagnosis of FLLs are of

great significance for providing appropriate treatment options and predicting prognosis.

Contrast-enhanced computed tomography (CECT) imaging is recommended by the international and national societies of hepatology as a first-line diagnostic tool for FLLs, because the vascularity and contrast-enhancing patterns of lesions provide useful information for diagnostic evaluation[3–5]. However, image-based diagnosis remains challenging due to the diversity of liver masses and complex imaging characteristics of liver lesions. Furthermore, interpretation of medical images is often subjective and influenced by the

A full list of affiliations appears at the end of the paper. ✉e-mail: xxu@bwh.harvard.edu; l.jiao@imperial.ac.uk; risheng-yu@zju.edu.cn; josephlau@cuhk.edu.hk; yizhouy@acm.org; srrsh_cxj@zju.edu.cn

experience and personal biases of radiologists. It has been reported that radiologists need to interpret each image in an average of 3-4 seconds in an 8-hour workday. With such a heavy workload, mistakes are inevitable due to fatigue. What's more, medical imaging data has been growing disproportionately compared to the number of well-trained radiologists. Therefore, an automated liver CT diagnosis system is urgently needed to improve the diagnostic accuracy and clinical efficiency.

With the recent progress of artificial intelligence (AI), especially the great successes of convolutional neural networks (CNNs) based deep learning[6,7], AI has been applied to various medical image analysis tasks with performance comparable to clinical experts, such as pulmonary disease identification[8], diabetic retinopathy identification and grading[9], skin lesion classification[10], distinguishing benign lesions from malignant ones, including breast cancer[11], lung cancer[12,13], and renal cancers[14], and so on.

Research on the diagnosis of liver cancers based on CT images has also made great progress. Yasaka et al. reported that in a study involving 580 patients, the area under the receiver operating characteristic curve (AUC) was 0.92 for the diagnosis of malignancy and the average accuracy of 0.84 for five-category classification[15]. Liang et al. achieved an average accuracy of 0.909 for four-category classification over 480 CT scans[16]. Shi et al. reported an AUC of 0.925 and an accuracy of 0.833 for HCC diagnosis in a study of 449 patients[17]. Cao et al. showed an average accuracy of 0.813 for four-category classification with 517 lesions[18]. Recent work by Zhou et al. reported that their method achieved an average accuracy of 0.734 for six-category classification and 0.825 for binary classification in 435 patients with 616 liver lesions[19].

However, even with such encouraging results, existing work still cannot be integrated into diagnostic workflows in clinical practice. First, most of the existing methods[15–18] still require radiologists to perform manual lesion extraction prior to lesion classification, thus failing to achieve an end-to-end fully automated diagnostic solution. Second, diagnostic analysis in existing work[15–19] only relies on image information, which does not conform to the actual clinical diagnostic workflow. Clinicians often combine clinical information with medical images to make accurate and comprehensive diagnoses. Most importantly, all existing studies involved very small sample sizes and few lesion types, and data sources were too limited to cover a variety of CT imaging devices from different manufacturers and data distributions. In other words, existing work has not been vigorously validated,

therefore, its robustness, reproducibility, and generalization ability are questionable for actual clinical practice.

To address the shortcomings of existing work, we retrospectively and prospectively collected large-scale data from 12,610 patients in 18 hospitals to train and validate a fully automated diagnostic system named Liver Artificial Intelligence Diagnosis System (LiAIDS). To our knowledge, this is the largest study ever undertaken, covering CT imaging devices from all mainstream manufacturers worldwide. More importantly, the proposed LiAIDS can robustly and accurately detect and differentiate lesions in a fully automated manner on the basis of contrast-enhanced CT scans and clinical information.

The architecture of the proposed LiAIDS consists of three main modules, namely lesion detection, liver segmentation, and lesion classification modules (as shown in Fig. 1). The lesion detection module was designed to automatically identify and localize all potential FLL candidates. The liver segmentation module serves as a false positive detector, filtering out lesions detected outside the liver region. Finally, the lesion classification module aims to differentiate the detected lesions into one of the seven most common disease types (i.e., HCC, ICC, HM, FNH, HH, HC, and HA) and further classify them as malignant or benign.

## Results

### LiAIDS development and primary validation

There were 12,610 patients collected. 11,385 participants from 18 hospitals in China between January 1st 2010 and June 30th, 2020 formed the retrospective cohort, and 1225 participants admitted to Sir Run Run Shaw Hospital (SRRSH) between July 1st 2020 and June 30th 2021 formed the prospective validation cohort. After conducting data quality control, 654 participants were excluded from the retrospective dataset. This resulted in 10,731 remaining participants, which were then divided into an internal cohort comprising data from 15 hospitals and three independent external cohorts, consisting of data from the remaining 3 hospitals, respectively. For the purpose of external validation, we specifically selected the three largest sites, excluding the largest SRRSH, to form our external validation cohorts. This approach guarantees a comprehensive performance assessment over a wide range of data, thereby capturing a greater diversity of patient scenarios. To be more specific, the first independent external validation cohort included 844 participants with 1623 FLLs from Zhangzhou Hospital, including 159 patients with malignant FLLs, 457 patients with benign FLLs, and 228 patients without FLLs. The second external

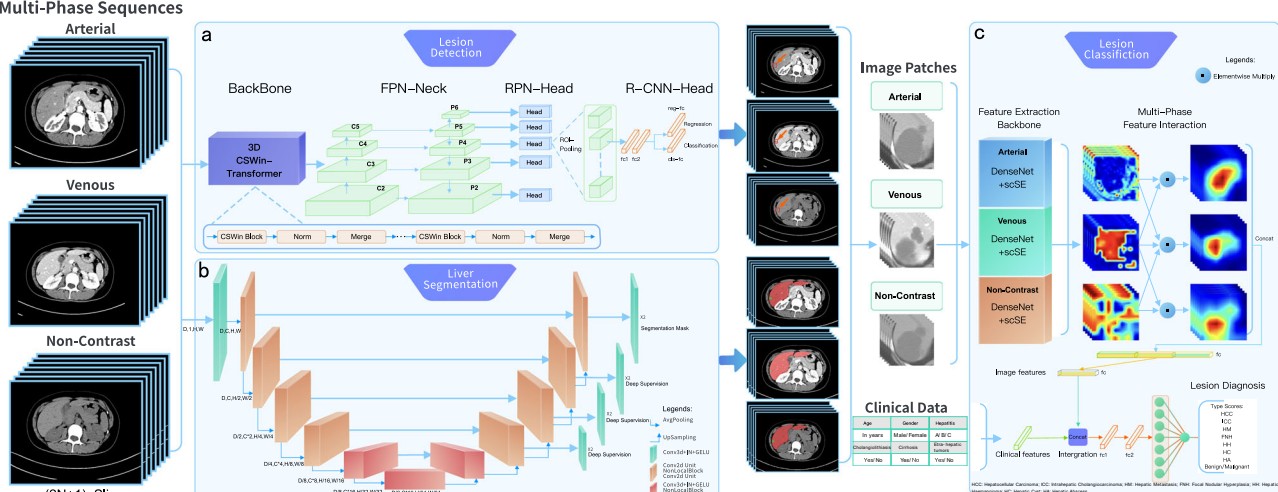

**Fig. 1 | Overview of the Liver Artificial Intelligence Diagnosis System (LiAIDS) proposed in this study. a** Lesion detection module, a faster-RCNN framework with newly extended 3D CSwin Transformer as feature extraction backbone; **b** Liver segmentation module, a hybrid convolutional network structure dominated by 2D convolution and supplemented by 3D convolution; **c** Lesion classification module utilizing features extracted from multi-phase images and clinical information.

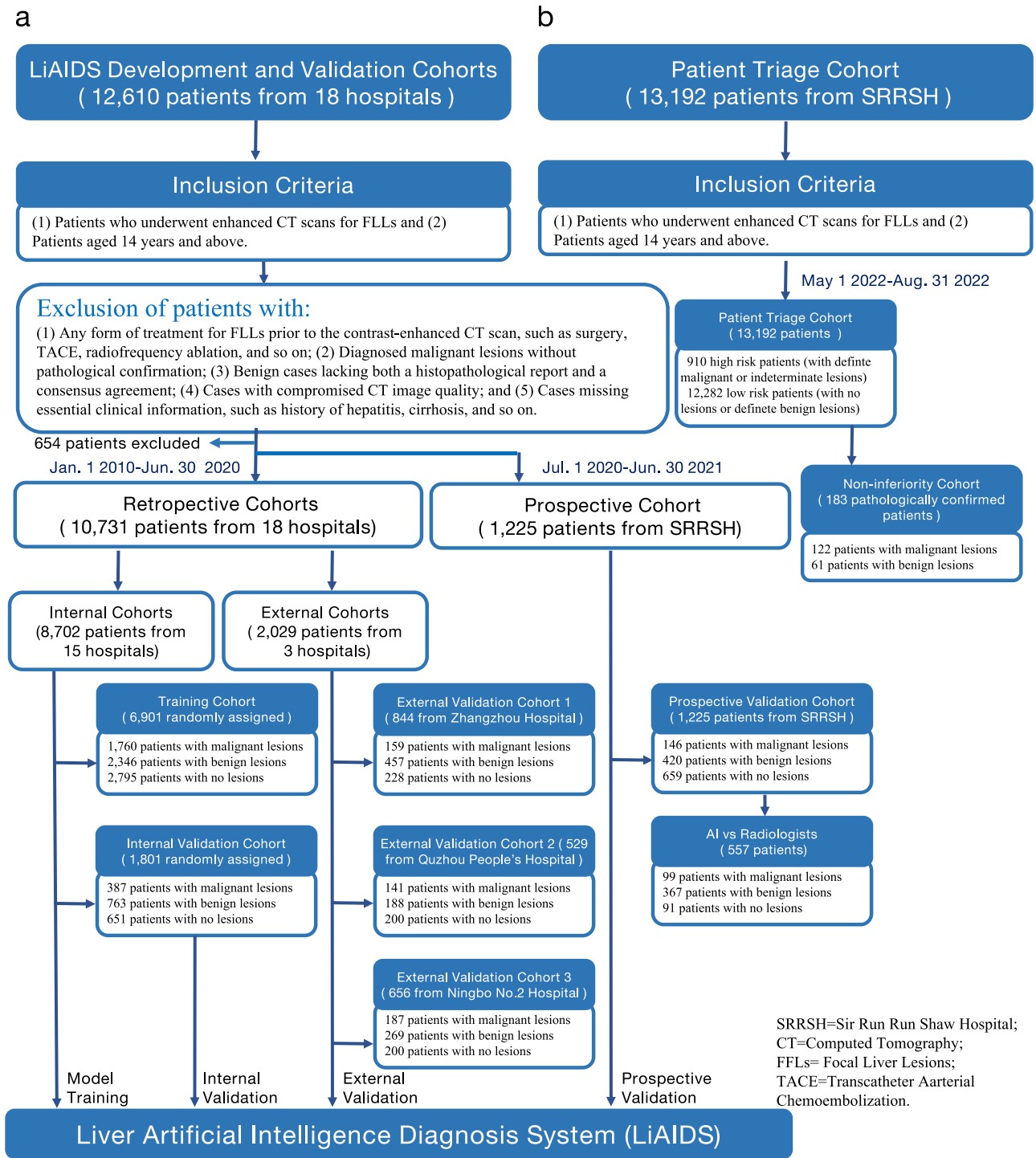

**Fig. 2 | The Flowchart of the Cohort Setup. a** Description of the training, internal, external, and prospective validation cohorts; **b** Cohort description of patient triage and non-inferiority trial.

validation cohort contained 529 participants with 1521 FLLs from Quzhou Hospital, including 141 patients with malignant FLLs, 188 patients with benign FLLs, and 200 patients without FLLs. The third external validation cohort had 656 participants with 1678 FLLs from Hwa Mei Ningbo No.2 Hospital, including 187 patients with malignant FLLs, 269 patients with benign FLLs, and 200 patients without FLLs. The prospective validation cohort consisted of 1,225 participants with 1804 FLLs, including 146 patients with malignant FLLs, 420 patients with benign FLLs, and 659 patients without FLLs. (Refer to Fig. 2 for more details).

Participants in the internal cohort were further randomly divided into internal training and validation cohorts at a 4:1 ratio. More specifically, the internal training cohort consisted of 6,901 participants with 12,564 FLLs, including 1760 patients with malignant FLLs, 2346 patients with benign FLLs, and 2795 patients without FLLs, while the internal validation cohort was composed of 1801 participants with 2855 FLLs, including 387 patients with malignant FLLs, 763 patients with benign FLLs, and 651 patients without FLLs. (Refer to Table 1 for comprehensive patient characteristics, the distribution of different lesion types and lesion sizes across all data cohorts).

**Table 1 | Baseline characteristics**

| | | Internal cohort from 15 hospitals (8702) | | External test cohorts (2029) | | | Prospective cohort(1225) |
|---|---|---|---|---|---|---|---|
| | | Training (6901) | Internal (1801) | ZZH (844) | QZH (529) | NBH (656) | SRRSH (1225) |
| Age, Years (Mean ± Std) | | 52.68 ±14.64 | 52.66 ±14.69 | 54.14 ±13.85 | 56.09 ±13.04 | 56.11 ±12.74 | 50.6 ±15.46 |
| Sex (Male / Female) | | 3807/3,094 | 1132/669 | 525/319 | 293/236 | 384/272 | 692/533 |
| History of hepatitis (No/Yes) | | 5650/1,251 | 1210/591 | 729/115 | 424/105 | 536/120 | 1145/80 |
| History of extra-hepatic tumors (No/Yes) | | 6069/832 | 1679/122 | 558/286 | 516/13 | 567/89 | 1088/137 |
| History of liver cirrhosis (No/Yes) | | 6329/572 | 1581/220 | 805/39 | 473/56 | 603/53 | 1185/40 |
| Number of patients | Malignant | 1760 | 387 | 159 | 141 | 187 | 146 |
| | Benign | 2346 | 763 | 457 | 188 | 269 | 420 |
| | No lesion | 2795 | 651 | 228 | 200 | 200 | 659 |
| Lesion numbers /size (Mean ± Std) (cm$^3$) | Malignant | 3914/117.1 ± 254.6 | 1155/80.8 ± 195.2 | 298/117.9 ± 303.4 | 171/105.7 ± 202.6 | 577/62.1 ± 172.5 | 437/43.8 ± 157.0 |
| | HCC | 1155/190.8 ± 375.6 | 752/157.7 ± 289.6 | 106/213.9 ± 288.3 | 106/129.9 ± 241.2 | 117/95.7 ± 168.0 | 64/129.3 ± 317.6 |
| | ICC | 613/118.7 ± 170.8 | 78/124.8 ± 168.6 | 13/268.2 ± 225.3 | 44/72.5 ± 87.3 | 28/122.4 ± 148.2 | 57/111.7 ± 167.5 |
| | HM | 2146/44.3 ± 127.6 | 325/18.7 ± 59.9 | 179/50.6 ± 297.4 | 21/25.5 ± 23.6 | 432/33.1 ± 171.9 | 316/14.4 ± 83.5 |
| | Benign | 8650/101.0 ± 319.5 | 1700/53.8 ± 118.9 | 1325/97.4 ± 279.4 | 1350/34.4 ± 116.6 | 1101/26.3 ± 83.5 | 1367/23.6 ± 72.5 |
| | HH | 2652/166.4 ± 566.1 | 304/61.1 ± 100.5 | 213/31.3 ± 92.3 | 211/76.7 ± 106.2 | 441/31.3 ± 95.5 | 473/19.8 ± 46.8 |
| | FNH | 453/31.9 ± 56.3 | 183/46.1 ± 66.4 | 42/9.9 ± 11.4 | 19/42.4 ± 34.0 | 18/37.8 ± 37.2 | 102/34.8 ± 139.5 |
| | HC | 5035/44.8 ± 142.2 | 1153/12.8 ± 109.2 | 801/70.2 ± 325.2 | 1118/25.4 ± 117.1 | 597/2.3 ± 6.7 | 721/12.7 ± 38.6 |
| | HA | 510/161.8 ± 198.6 | 60/176.5 ± 189.7 | 269/235.8 ± 235.1 | 2/295.6 ± 3.4 | 45/109.9 ± 139.3 | 71/108.6 ± 151.0 |
| Size (cm$^3$) (Mean ± Std) | | 108.0 ± 293.1 | 64.9 ± 155.5 | 101.8 ± 284.9 | 42.9 ± 131.9 | 40.0 ± 126.7 | 28.8 ± 101.3 |
| Lesions/patient (Mean ± Std) | | 1.82 ± 3.51 | 1.59 ± 2.94 | 1.92 ± 3.75 | 2.88 ± 5.45 | 2.56 ± 5.15 | 1.47 ± 3.47 |

*Std* Standard Deviation, *ZZH* Zhangzhou Hospital, *QZH* Quzhou People's Hospital, *NBH* Ningbo No.2 Hospital.

In this study, all modules of LiAIDS, including lesion detection, liver segmentation, and lesion classification, were independently trained using the internal training cohort. The model performance of LiAIDS was retrospectively validated by the internal and three independent external validation cohorts, and further prospectively validated by the prospective validation cohort.

In the retrospective study, LiAIDS consistently achieved high AUC performance for the seven-category classification with a macro-average AUC of 0.982 (95% CI: 0.974–0.989) on the internal validation cohort, and a macro-average AUC of 0.973 (95% CI: 0.960–0.985), 0.970 (95% CI: 0.958–0.981), and 0.958 (95% CI: 0.929–0.980), on the three independent external validation cohorts, respectively. The AUC for the binary classification was also consistently high with 0.980 (95% CI: 0.967–0.990) on the internal validation cohort, and 0.989 (95% CI: 0.983–0.994), 0.977 (95% CI: 0.963–0.988), and 0.981 (95% CI: 0.972–0.989) on the three independent external validation cohorts, respectively. (More details can be found in Fig. 3a–d).

In the prospective study, LiAIDS achieved a macro-average AUC of 0.967 (95% CI: 0.956-0.977) for the seven-category classification and an AUC of 0.971 (95% CI: 0.957–0.981) for the binary classification, respectively (Fig. 3e).

The image feature representations learned by LiAIDS were also visualized using t-SNE, with individual points representing sample lesions from the prospective validation cohort. Each point is a two-dimensional projection of 588-dimensional feature vector generated by the LiAIDS lesion classification module. As shown in Fig. 3f, we observed distinct clustering of points representing the four types of benign lesions, which were clearly separated from malignant FLLs, particularly HCC and HM clusters. Within malignant FLLs, points representing HCC lesions were easily discernible from those representing HM lesions. However, there was a tendency for points representing ICC lesions to be confused with the other two types of malignant lesions. It is also noteworthy that there was some degree of overlap between the benign clusters of FNH and HH, as consistently reflected in the confusion matrices (Fig. 3h–l), where FNH and HH exhibited a higher probability of being misclassified as each other. We

undertook an in-depth analysis of those FNH and HH samples that exhibited intertwined characteristics. Remarkably, both categories demonstrated significantly smaller mean volumes (cm$^3$) compared to the overall volume distribution within their respective data cohort (FNH: 6.309 ± 5.72 vs 34.8 ± 139.5 and HH: 2.457 ± 2.36 vs 19.8 ± 46.8). In a clinical setting, diagnosing of FNH and/or HH can be challenging, particularly in the case of smaller lesions. The usual procedure for managing such cases typically involves additional imaging studies, predominantly MRI, or ongoing clinical assessments. Given the benign nature of these conditions, immediate differentiation is not vital. However, diligent and continued monitoring of these conditions is essential due to potential alterations and progression over time. LiAIDS also demonstrated high diagnostic performance in terms of accuracy, sensitivity, specificity, and precision both retrospectively and prospectively on all five validation cohorts. For example, on the internal validation cohort, LiAIDS achieved an accuracy of 0.934 (95% CI: 0.911-0.955), a sensitivity of 0.922 (95% CI: 0.879–0.958), a specificity of 0.943 (95% CI: 0.912–0.969), and a precision of 0.922 (95% CI: 0.881–0.961) for the binary classification. The binary classification performance of LiAIDS across all validation cohorts can be found in Fig. 3g, where the distribution of probability scores among different cohorts provides additional insights into the model's performance about its generalizability. A consistent distribution observed across cohorts indicates a high level of performance consistency, demonstrating robust generalizability. The confusion matrices in Fig. 3h–l also provide a comprehensive understanding of the model's performance across different lesion types. It is noteworthy that there were only 2 HA lesions in QZH and neither of them were correctly identified as shown in Fig. 3 j, resulting in a lower confidence interval for HA in Fig. 3c and Table S1. (For more detailed diagnostic performance of LiAIDS, please refer to Table 2).

**Performance comparison with radiologists**
To clearly understand the performance of LiAIDS against radiologists in a clinical setting, we compared the performance of LiAIDS and six general radiologists with different levels of expertise. The six general

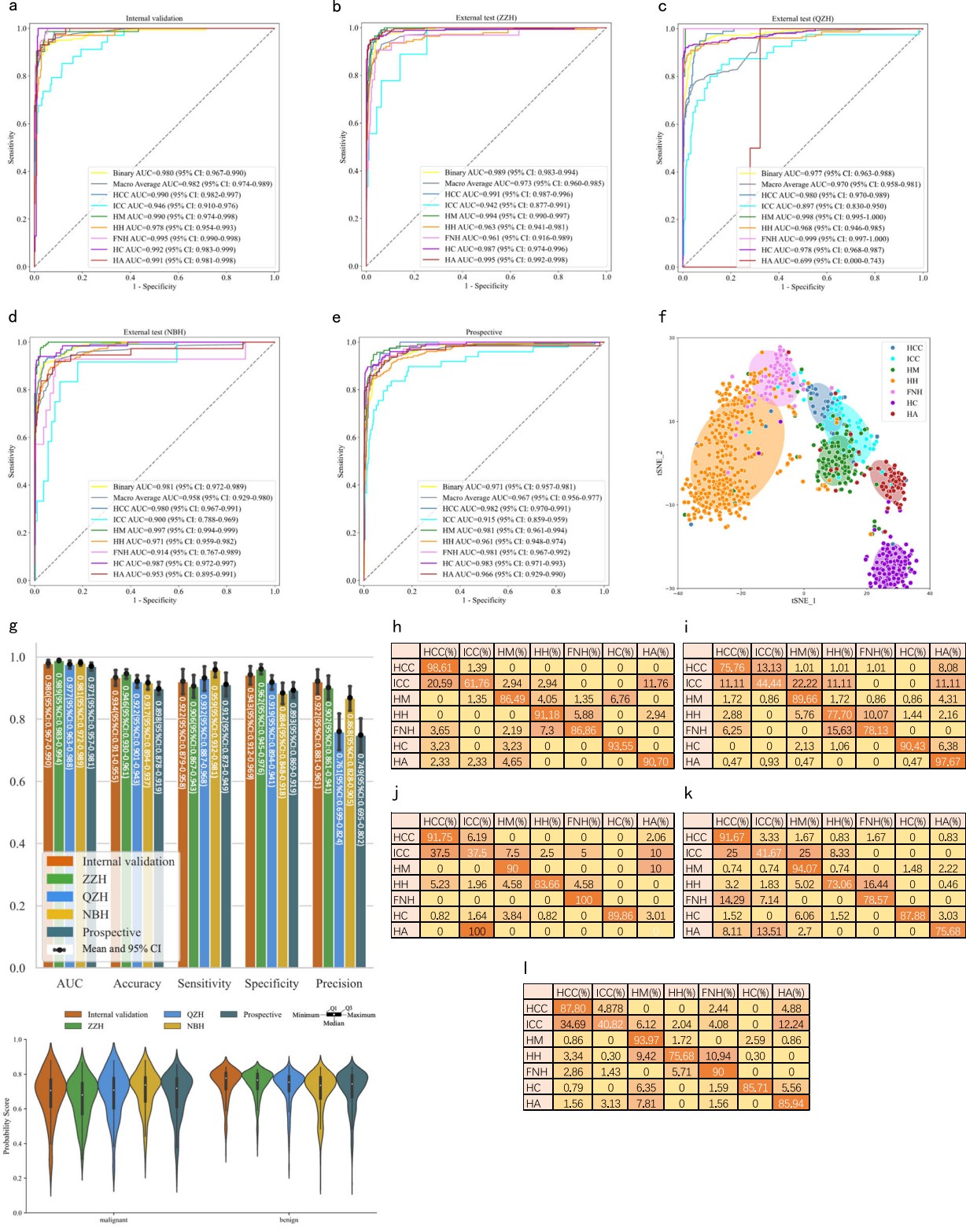

AUC=Area Under Curve; 95%CI = 95% Confidence Interval; ZZH=Zhangzhou Hospital; QZH=Quzhou People's Hospital; NBH=Ningbo No.2 Hospital; HCC=Hepatocellular Carcinoma; ICC=Intrahepatic Cholangiocarcinoma; HM=Hepatic Metastasis; HC=Hepatic Cyst; HH=Hepatic Haemangioma; FNH=Focal Nodular Hyperplasia; HA=Hepatic Abscess; tSNE=t-Distributed Stochastic Neighbor Embedding; Q1=Lower Quartile; Q3=Upper Quartile.

**Fig. 3 | Diagnostic Performance of the proposed Liver Artificial Intelligence Diagnosis System (LiAIDS) in All Cohorts. a** ROC curve of the internal validation cohort; **b** ROC curve of the external validation cohort of ZZH; **c** ROC curve of the external validation cohort of QZH; **d** ROC curve of the external validation cohort of NBH; **e** ROC curve of the prospective validation cohort of SRRSH; **f** t-SNE plot of the prospective validation cohort. Scatter plot illustrating the clustering of lesion images in the prospective cohort. Each point represents an image of a lesion, and the color indicates its true class; **g** Performance of the binary classification across all validation cohorts, evaluated using all five metrics; **h** Confusion matrix of the internal validation cohort; **i** Confusion matrix of the external validation cohort of ZZH; **j** Confusion matrix of the external validation cohort of QZH; **k** confusion matrix of the external validation cohort of NBH; **l** Confusion matrix of the prospective validation cohort of SRRSH's. Source data are provided as a Source Data file (Source_data_Figure_3.xlsx).

**Table 2 | Diagnostic performance of LiAIDS on different cohorts**

| | | Internal validation cohort | External test cohorts | | | Prospective cohort |
|---|---|---|---|---|---|---|
| | | | ZZH | QZH | NBH | |
| Binary | | | | | | |
| AUC (95%CI) | | 0.980 (0.967–0.990) | 0.989 (0.983–0.994) | 0.977 (0.963–0.988) | 0.981 (0.972–0.989) | 0.971 (0.957–0.981) |
| Accuracy (95%CI) | | 0.934 (0.911–0.955) | 0.946 (0.930–0.961) | 0.922 (0.901–0.943) | 0.917 (0.894–0.937) | 0.898 (0.878–0.919) |
| Precision (95%CI) | | 0.922 (0.881–0.961) | 0.902 (0.861–0.941) | 0.761 (0.699–0.824) | 0.868 (0.828–0.905) | 0.749 (0.695–0.802) |
| Sensitivity (95%CI) | | 0.922 (0.879–0.958) | 0.906 (0.867–0.943) | 0.932 (0.887–0.968) | 0.959 (0.932–0.981) | 0.912 (0.873–0.949) |
| Specificity (95%CI) | | 0.943 (0.912–0.969) | 0.962 (0.945–0.976) | 0.919 (0.894–0.941) | 0.884 (0.848–0.918) | 0.893 (0.869–0.919) |
| Seven-category | | | | | | |
| AUC (95%CI) | | 0.982 (0.974–0.989) | 0.973 (0.960–0.985) | 0.970 (0.958–0.981) | 0.958 (0.929–0.980) | 0.967 (0.956–0.977) |
| Accuracy (95%CI) | | 0.880 (0.847–0.911) | 0.872 (0.848–0.895) | 0.856 (0.831–0.881) | 0.828 (0.798–0.859) | 0.805 (0.777–0.833) |
| Patient-wise Accuracy (95%CI) | | | | | | |
| Binary | | 0.951 (0.925, 0.974) | 0.943 (0.923, 0.961) | 0.889 (0.855, 0.923) | 0.909 (0.878, 0.937) | 0.873 (0.842–0.903) |
| Seven-category | | 0.912 (0.879, 0.941) | 0.904 (0.878, 0.928) | 0.886 (0.848, 0.919) | 0.884 (0.850, 0.915) | 0.833 (0.799–0.864) |
| CT acquisition parameters Seven-category Accuracy (95%CI) | | | | | | |
| Slice thickness | [2.5–5 mm) | 0.729 (0.665, 0.792) | – | – | – | – |
| | [5.0–7 mm) | 0.892 (0.859, 0.925) | 0.872 (0.848–0.895) | 0.856 (0.831–0.881) | 0.828 (0.798–0.859) | 0.789 (0.760, 0.818) |
| | [7.0–10 mm) | 0.835 (0.758, 0.912) | – | – | – | 0.949 (0.899, 0.987) |
| mAs | [0–100) | 0.859 (0.809, 0.905) | 0.740 (0.630, 0.836) | 0.849 (0.813, 0.881) | 0.837 (0.800, 0.872) | 0.813 (0.776, 0.853) |
| | [100,200] | 0.891 (0.842, 0.939) | 0.870 (0.832, 0.905) | 0.861 (0.799, 0.917) | 0.800 (0.746, 0.859) | 0.825 (0.780, 0.867) |
| | [200,500) | 0.918 (0.852, 0.984) | 0.897 (0.868, 0.927) | 0.878 (0.800, 0.944) | 0.917 (0.750, 1.000) | 0.722 (0.639, 0.796) |

*LiAIDS* Liver Artificial Intelligence Diagnosis System, *AUC* area under the receiver operating characteristic curve, *ZZH* Zhangzhou Hospital, *QZH* Quzhou People's Hospital, *NBH* Ningbo No. 2 Hospital.

radiologists were divided into two groups, a junior group consisted of 3 radiologists with 5–10 years of experience in abdominal imaging diagnosis, and a senior group consisted of 3 radiologists with 10–20 years of experience in abdominal imaging diagnosis. In this study, 557 patients were randomly selected from the prospective validation cohort for performance comparison, including 99 patients with malignant lesions, 367 patients with benign lesions, and 91 patients without lesions.

To comprehensively evaluate the clinical value of LiAIDS, this study conducted performance comparisons between radiologists and AI, as well as collaborations between radiologists and AI. Specifically, LiAIDS was compared with the diagnoses made by radiologists both with and without the assistance of LiAIDS. To ensure a fair comparison, the radiologists were blinded to the histopathological and radiological reports contained in the electronic health records of the patients under study. They only relied on the CT scans and clinical information of the patients to make independent diagnoses, either with or without the guidance of LiAIDS. Importantly, it is worth noting that there was a minimum one-month interval between the radiologists' diagnostic sessions with and without the utilization of LiAIDS.

The comparison results presented in Fig. 4a and b revealed that LiAIDS performed better than the group of junior radiologists and demonstrated a performance on par with the group of senior radiologists. In particular, LiAIDS achieved F1-scores of 0.940 for benign lesions and 0.692 for malignant ones, while the F1-scores for the junior group were 0.830, 0.890, and 0.860 for benign lesions, and 0.230, 0.360, and 0.330 for malignant cases. In comparison, the senior group recorded F1-scores of 0.950, 0.920, and 0.950 for benign lesions, and 0.620, 0.550, and 0.650 for malignant cases. As expected, the performance of all radiologists improved with the assistance of LiAIDS. Consequently, the three junior radiologists reached performance comparable to those achieved solely by LiAIDS, while the three senior radiologists attained performance that was either superior to or on par with LiAIDS when aided by it. For instance, the F1-scores of the three junior radiologists for benign lesions rose from 0.830, 0.890, and 0.860 to 0.936, 0.946, and 0.944, respectively. Likewise, for malignant cases, the F1-scores improved from 0.230, 0.360, and 0.330 to 0.680,

0.671, and 0.667 respectively. The performance of the senior radiologists also exhibited improvements, with F1-scores for benign lesions rising from 0.950, 0.920, and 0.950 to 0.961, 0.950, and 0.958 respectively, and for malignant cases, from 0.620, 0.550, and 0.650 to 0.753, 0.679, and 0.727 respectively. Notably, junior radiologist 1, who had the lowest performance, saw the most substantial improvement. The sensitivity, specificity, and F1-score for benign lesions improved from 0.767, 0.361, and 0.830 to 0.899, 0.854, and 0.936 respectively. For malignant cases, these metrics increased from 0.361, 0.767, and 0.230 to 0.854, 0.899, and 0.680 respectively. The rationale behind this could be attributed to the heavy reliance on LiAIDS. Additionally, the ROC curve depicted in Fig. 4c highlighted that LiAIDS surpassed all radiologists in binary classification, achieving an AUC of 0.968 (95% CI: 0.944-0.986). Importantly, junior radiologists saw a performance boost when assisted by LiAIDS. In Fig. 4d, we compared lesion detection performance between LiAIDS and the six radiologists. LiAIDS achieved a recall rate of 0.965 (95% CI: 0.960-0.970), higher than both the junior group (recall rates: 0.766, 0.883, 0.796) and the senior group (recall rates: 0.911, 0.911, 0.962) (all p < 0.001). With LiAIDS support, the junior radiologists' recall rates increased (from 0.766, 0.883, 0.796 to 0.970, 0.954, 0.965), achieving levels comparable to LiAIDS's recall rate of 0.965. Meanwhile, the senior group's detection performance also improved (from 0.911, 0.911, 0.962 to 0.975, 0.969, 0.972), surpassing the performance of LiAIDS. (Please refer to Table 3 and Table S2 for more detailed information).

### Comprehensive performance analysis on multi-modal vs single-modal

As previously mentioned, the integration of clinical information alongside image characteristics is crucial for accurate diagnosis. In this study, clinical information included both essential patient information (age and gender) and pertinent medical history, including hepatitis, cirrhosis, cholangiolithiasis, and extra-hepatic tumors. The performance of LiAIDS was then compared with models trained solely on clinical information or image data. The comparison results demonstrated that LiAIDS outperformed models that were trained exclusively on either clinical information or image data in all five validation

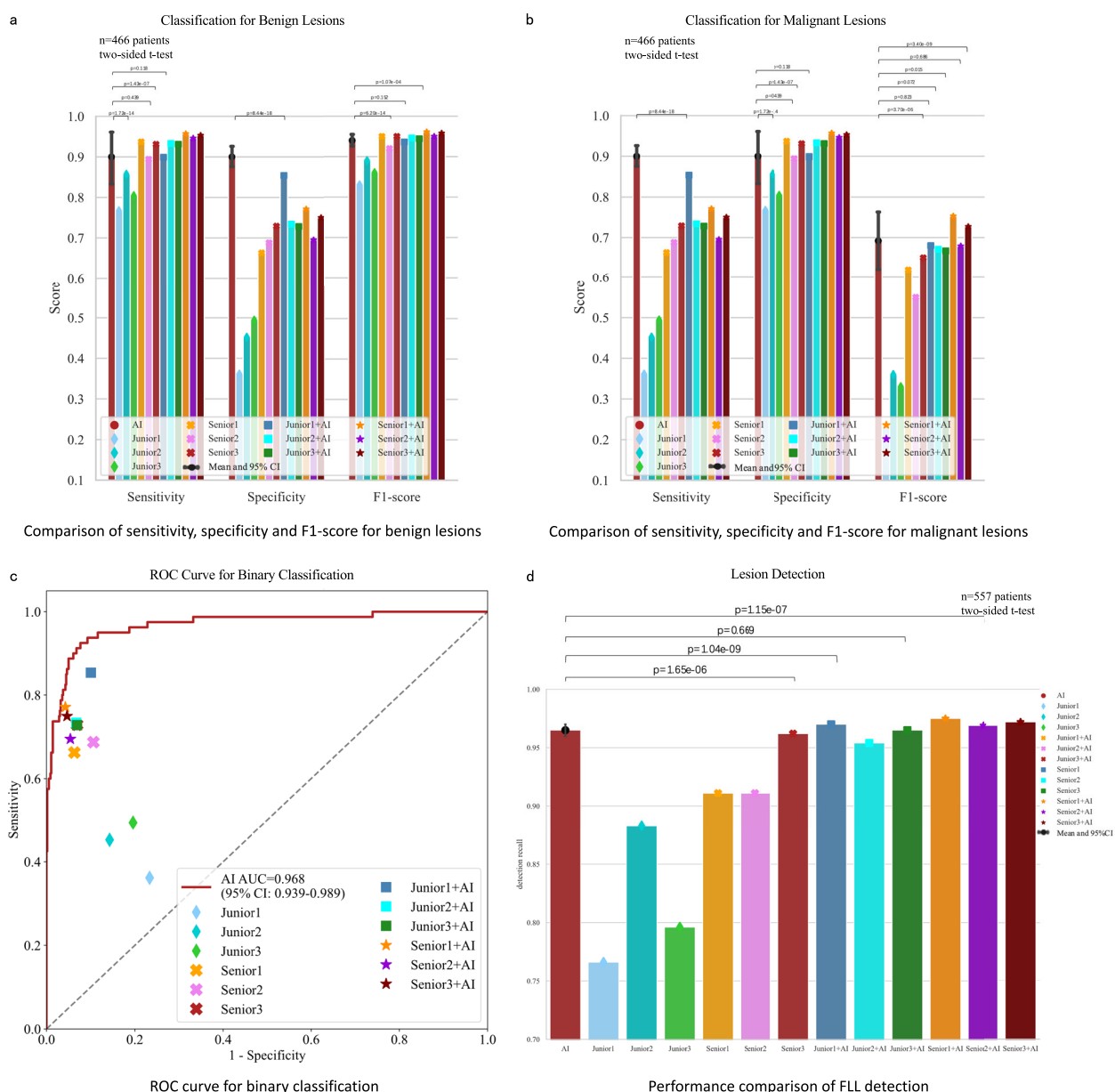

AI=Artificial Intelligence; 95%CI = 95% Confidence Interval; ROC=Receiver Operating Characteristic; AUC=Area Under Curve; FLL=Focal Liver Lesions.

**Fig. 4 | Performance Comparison of LiAIDS, Practicing Radiologists with and without the assistance of LiAIDS. a** Comparison of sensitivity, specificity and F1-score for benign lesions; **b** Comparison of sensitivity, specificity and F1-score for malignant lesions; **c** ROC curve for binary classification; **d** Performance comparison of FLL detection. Source data are provided as a Source Data file (Source_data_Figure_4.xlsx).

cohorts across the metrics of AUC, accuracy, sensitivity, and precision, as shown in Fig. 5a–e. For instance, in the seven-category classification, LiAIDS achieved accuracy scores of 0.880 (95% CI: 0.847-0.911), 0.872 (95% CI: 0.848-0.895), 0.856 (95% CI: 0.831-0.881), 0.828 (95% CI: 0.798-0.859), and 0.805 (95% CI: 0.777-0.833) in the internal, three external, and prospective cohorts, respectively. In contrast, the model trained solely with clinical data yielded accuracy scores of 0.649 (95% CI: 0.605-0.694), 0.484 (95% CI: 0.450-0.519), 0.167 (95% CI: 0.140-0.194), 0.489 (95% CI: 0.449-0.529), and 0.318 (95% CI: 0.287-0.351). The model trained solely on image data produced accuracy scores of 0.786 (95% CI: 0.748-0.824), 0.794 (95% CI: 0.766-0.822), 0.809 (95% CI: 0.778-0.839), 0.701 (95% CI: 0.665-0.736), and 0.768 (95% CI: 0.740-0.797). Regarding specificity, we noted that the model utilizing only clinical data achieved the highest score of 0.980 (95% CI: 0.976-0.985) in the NBH cohort, while LiAIDS and the image data-based model

exhibited scores of 0.969 (95% CI: 0.964-0.975) and 0.949 (95% CI: 0.942-0.955) respectively. Nonetheless, in the remaining four cohorts, LiAIDS surpassed models reliant solely on either clinical information or image data. In general, LiAIDS exhibited a macro-average AUC of 0.982 (95% CI: 0.974-0.989) for the seven-category classification and an AUC of 0.980 (95% CI: 0.967-0.990) for the binary classification. In contrast, the model utilizing only image data achieved a lower macro-average AUC of 0.949 (95% CI: 0.938-0.960) for the seven-category classification and an AUC of 0.956 (95% CI: 0.935-0.974) for the binary classification. The model using solely clinical data yielded an even lower macro-average AUC of 0.801 (95% CI: 0.776-0.827) for the seven-category classification and an AUC of 0.927 (95% CI: 0.899-0.952) for the binary classification as depicted in Fig. 5f in the internal validation cohort. These results highlight the improvement in FLL classification achieved by incorporating both CECT images and clinical information

**Table 3 | Performance comparison between LiAIDS and radiologists on detection and diagnosis of FLLs**

| Approach | Recall of detection | Accuracy | Benign lesions | | | Malignant lesions | | |
|---|---|---|---|---|---|---|---|---|
| | | | Sen | Spe | F1-score | Sen | Spe | F1-score |
| Junior1 | 0.766 | 0.727 | 0.767 | 0.361 | 0.830 | 0.361 | 0.767 | 0.230 |
| Junior2 | 0.883 | 0.810 | 0.857 | 0.452 | 0.890 | 0.452 | 0.857 | 0.360 |
| Junior3 | 0.796 | 0.767 | 0.804 | 0.494 | 0.860 | 0.494 | 0.804 | 0.330 |
| Senior1 | 0.911 | 0.905 | 0.937 | 0.663 | 0.950 | 0.663 | 0.937 | 0.620 |
| Senior2 | 0.911 | 0.870 | 0.894 | 0.688 | 0.920 | 0.688 | 0.894 | 0.550 |
| Senior3 | 0.962 | 0.907 | 0.931 | 0.729 | 0.950 | 0.729 | 0.931 | 0.650 |
| Junior1+LiAIDS | 0.97 | 0.893 | 0.899 | 0.854 | 0.936 | 0.854 | 0.899 | 0.680 |
| Junior2+LiAIDS | 0.954 | 0.907 | 0.933 | 0.734 | 0.946 | 0.734 | 0.933 | 0.671 |
| Junior3+LiAIDS | 0.965 | 0.904 | 0.931 | 0.728 | 0.944 | 0.728 | 0.931 | 0.667 |
| Senior1+LiAIDS | 0.975 | 0.932 | 0.957 | 0.771 | 0.961 | 0.771 | 0.957 | 0.753 |
| Senior2+LiAIDS | 0.969 | 0.913 | 0.946 | 0.695 | 0.950 | 0.695 | 0.946 | 0.679 |
| Senior3+LiAIDS | 0.972 | 0.927 | 0.953 | 0.750 | 0.958 | 0.750 | 0.953 | 0.727 |
| LiAIDS | 0.965 | 0.900 | 0.900 | 0.900 | 0.940 | 0.900 | 0.900 | 0.692 |

*LiAIDS* Liver Artificial Intelligence Diagnosis System, *Spe* Specificity, *Sen* Sensitivity.

in LiAIDS. (For more detailed information, please refer to Fig. 5 and Table S3).

## Comprehensive performance analysis on multicenter vs single-center

To effectively demonstrate the efficacy of our large-scale multicenter study, we conducted additional experimental comparisons between multicenter and single-center training data. To ensure fairness, the sample sizes of the multicenter and single-center training datasets used in this comparison remained the same. The single-center training data consisted of 3,167 randomly sampled cases from the training set of SRRSH, while the multicenter training data comprised 3,167 cases randomly sampled from the original training dataset collected from 15 hospitals. As expected, the comparison results on the three independent external validation cohorts revealed that the model trained on multicenter data statistically outperformed or performed on par with the model trained on the single-center data across all metrics including AUC, accuracy, sensitivity, specificity, and precision. For instance, in the seven-category classification, the accuracy scores of the model trained on multicenter data were 0.891 (95% CI: 0.870-0.912), 0.826 (95% CI: 0.797-0.854), and 0.813 (95% CI: 0.779-0.844) on the external cohorts of ZZH, QZH, and NBH, respectively. In comparison, the accuracy scores of the model trained on single-center data were 0.846 (95% CI: 0.821-0.871), 0.810 (95% CI: 0.782-0.838), and 0.801(95% CI: 0.769-0.833) on the external cohorts of ZZH, QZH, and NBH, respectively. Importantly, there were statistically significant improvements in both sensitivity and specificity between models trained on multicenter and single-center data (p < 0.001). This finding highlights the robustness and generalization ability of LiAIDS across different medical centers, indicating its suitability for application in actual clinical practice. (Detailed information can be found in Fig. S1 and Table S4).

## Comprehensive performance analysis on lesion classification with vs without phase interaction module

For lesion classification using multi-phase CT images, existing methods[15–19] often performed multi-phase feature fusion by simply concatenating multi-phase features, ignoring the intrinsic correlation between different image phases. In this study, a phase interaction module was investigated and integrated into the lesion classification module of LiAIDS. The phase interaction module explicitly exploited the intrinsic correlation between different image phases. Ablation experiments performed on models with and without the phase

interaction module showed that the model with the phase interaction module outperformed the model without this module on all metrics, including AUC, accuracy, sensitivity, specificity, and precision. More specifically, for the seven-category classification, the accuracy of the model with the phase interaction module was 0.872(95% CI: 0.848-0.895), 0.856(95% CI: 0.831-0.881), 0.828(95% CI: 0.798-0.859) on the external cohorts of ZZH, QZH, and NBH, respectively, and the accuracy of the model without the phase interaction module was 0.868(95% CI: 0.845-0.891), 0.826(95% CI: 0.798-0.854), 0.801(95% CI: 0.769-0.834) on the external cohorts of ZZH, QZH, and NBH, respectively. (More details on the performance comparison can be found in Figure S2 and Table S5; more details on the phase interaction module can be found in Fig. 1 and Methods).

## Comprehensive performance analysis on lesion detection with 3D CSwin Transformer

To fully leverage the 3D contextual information embedded in CT slices, we made modifications to the state-of-the-art CSwin Transformer[20] in this study. The modified 3D CSwin Transformer was used as the feature extraction backbone within the faster-RCNN[21] framework for lesion detection. The original parameters of the faster-RCNN were retained in our approach. In our study, the correct detection of lesions was determined based on the overlap of bounding boxes between the detected lesions and the ground-truth bounding boxes. Specifically, we considered a detection to be correct if the Intersection over Union (IOU) value was greater than 0.3 and/or the Intersection over Minimum (IOM) value was greater than 0.5. These thresholds were used to assess the degree of overlap and ensure accurate detection. Ablation experiments on models with and without the modified 3D CSwin Transformer backbone showed that the model with the modified 3D backbone outperformed the model without it on all 5 validation datasets on all lesions (p < 0.001 on all cohorts). More specifically, the overall recall rate for all lesions was improved from 0.919 (95% CI: 0.905-0.933) to 0.930 (95% CI: 0.918-0.942) on the internal validation cohort, and from 0.957 (95% CI:0.952-0.962) to 0.963 (95% CI: 0.958-0.968) on the external ZZH cohort, and from 0.971 (95% CI: 0.966-0.977) to 0.973 (95% CI: 0.967-0.978) on the external QZH cohort, and from 0.924 (95% CI: 0.916-0.932) to 0.928 (95% CI: 0.920-0.936) on the external NBH cohort, and from 0.945 (95% CI: 0.937-0.953) to 0.951 (95% CI: 0.944-0.957) in the prospective cohort. Notably, for lesions smaller than 1 cm, the recall rate was improved from 0.862 (95% CI: 0.829-0.895) to 0.893 (95% CI: 0.861-0.925) in the internal validation cohort, from 0.907 (95% CI: 0.891-0.924) to 0.937 (95% CI: 0.923-0.951)

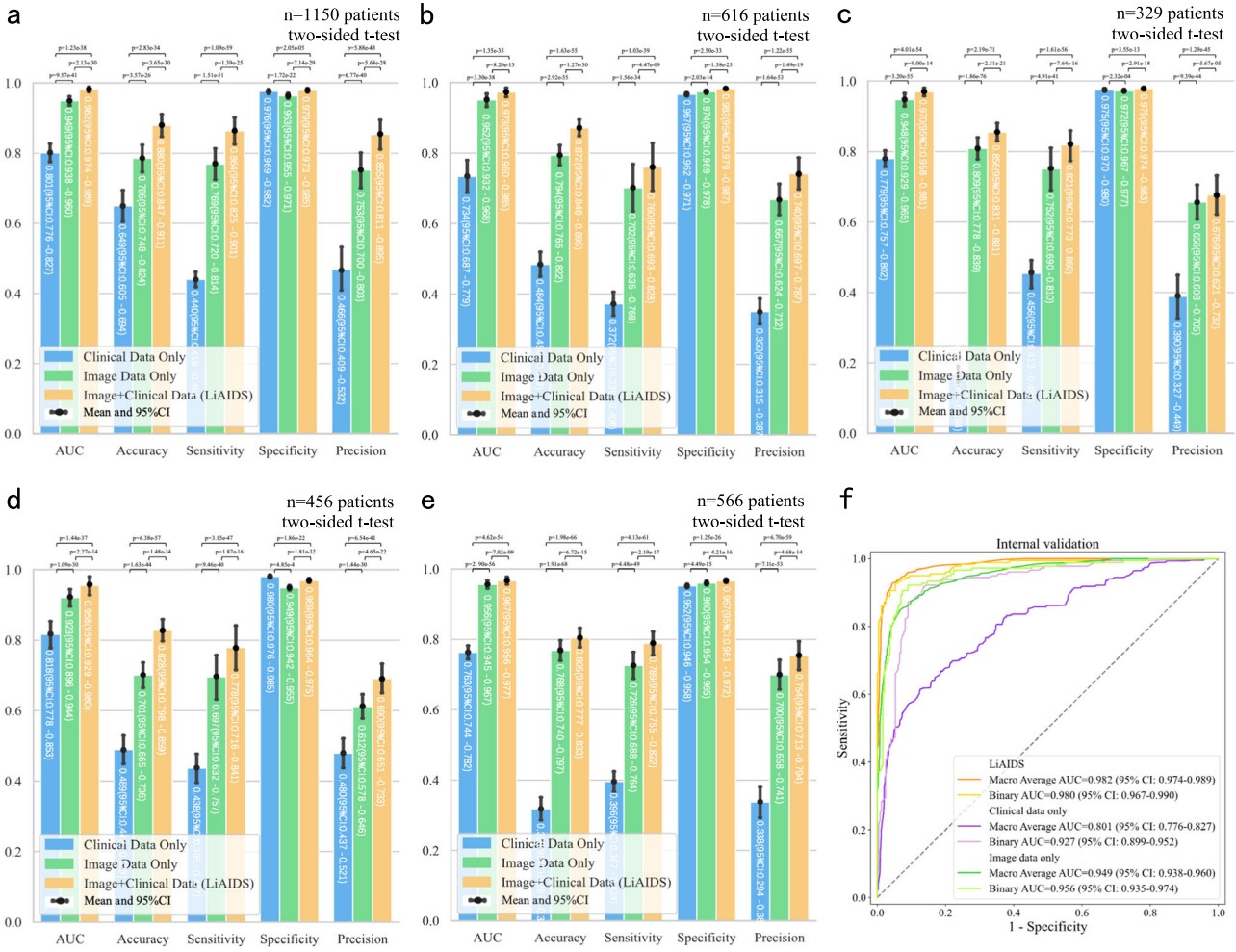

LiAIDS=Liver Artificial Intelligence Diagnosis System; 95%CI = 95% Confidence Interval; AUC=Area Under Curve.

**Fig. 5 | Comparison of Diagnostic Performance of Models Trained with Different Data Modalities (Single-modal vs Multi-modal) across All Cohorts.**
**a** Comparison of the seven-category classification in the internal validation cohort; **b** Comparison of the seven-category classification in the external validation cohort of ZZH; **c** Comparison of the seven-category classification in the external validation cohort of QZH; **d** Comparison of the seven-category classification in the external validation cohort of NBH; **e** Comparison of the seven-category classification in the prospective validation cohort of SRRSH; **f** ROC curves for models using different data modalities in the internal validation cohort. Source data are provided as a Source Data file (Source_data_Figure_5.xlsx).

in the external ZZH cohort, from 0.942 (95% CI: 0.927-0.958) to 0.952 (95% CI: 0.937-0.966) on the external QZH cohort, from 0.878 (95% CI: 0.851-0.905) to 0.898(95% CI:0.874-0.921) in the external NBH cohort, and from 0.922 (95%CI:0.901-0.943) to 0.948 (95%CI:0.931-0.964) in the prospective cohort. That is, for lesions smaller than 1 cm, the improvement in detection performance was greater. This feature has important clinical value because small lesions are easily missed in clinical practice and the detection of small lesions is challenging. Upon further examination of the missed lesions, it was discovered that all of them were benign. The concurrence between LiAIDS and the assessments made by junior and senior radiologists demonstrated high agreement, with measures of 0.9487 and 0.923, respectively. (More details on the performance comparison can be found in Fig. 6 and Table S6).

## Clinical application in patient triage

LiAIDS has showcased diagnostic capabilities in both retrospective and prospective settings. Moreover, during the inference phase, LiAIDS, on average, took 128 seconds to deliver a diagnosis upon receiving the CT images. This incorporated lesion detection, segmentation, and classification, all performed on a single workstation. This promising outcome inspired us to carry out a further study to investigate the

potential utility of LiAIDS in patient triage. As demonstrated in Fig. 7a, we collected data from 13,192 consecutive patients enrolled between May 1st, 2022 and August 31th, 2022 at SRRSH. In this investigation, LiAIDS automatically categorized 76.46% of the total patients as low risk (those with no lesions or definitive benign lesions), with the remaining 23.54% classified as high risk (those with definitive malignant lesions or indeterminate lesions). Upon a retrospective analysis of all cases by medical physicians, LiAIDS achieved a Negative Predictive Value (NPV) of 99.0% and a Positive Predictive Value (PPV) of 28.67%. Essentially, LiAIDS identified 76.46% of the patients as negative, with 99% of those predicted negative cases confirmed as true negatives. While there is no specific report on the current standard of care, a recent study on 214 patients[22] that concentrated on identifying malignancy in FLLs found that standard ultrasound had an NPV of 78% and a PPV of 60%. In contrast, CT scans yielded an NPV of 86% and a PPV of 81%, and MRI provided an NPV of 91% and a PPV of 68%. We must underscore that our goal is to reduce radiologists' workload by minimizing the number of cases they need to review, not to replace them with LiAIDS for diagnostic purposes. As such, a lower PPV is not a major concern in this context. The triage process using LiAIDS can decrease radiologists' workload by 76.46%, while resulting in less than 1% false negative cases.

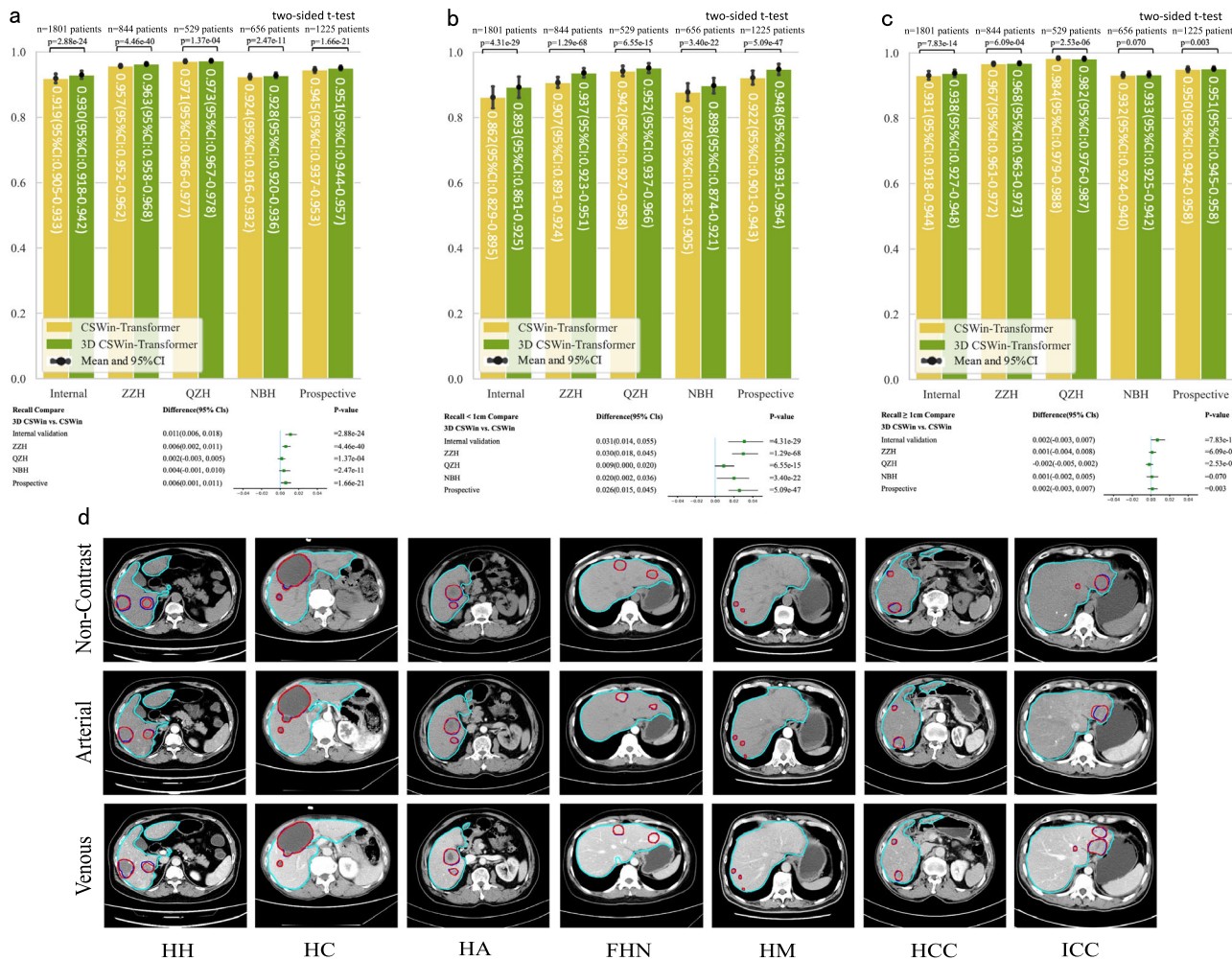

CSWin-Transformer=Cross-Shaped Window Transformer; ZZH=Zhangzhou Hospital; QZH=Quzhou People's Hospital; NBH=Ningbo No.2 Hospital; 95%CI = 95% Confidence Interval; HCC=Hepatocellular Carcinoma; ICC=Intrahepatic Cholangiocarcinoma; HM=Hepatic Metastasis; HC=Hepatic Cyst; HH=Hepatic Haemangioma; FNH=Focal Nodular Hyperplasia; HA=Hepatic Abscess.

**Fig. 6 | Comparison of Lesion Detection Performance of Different Models (Trained with and without 3D CSwin-transformer) and Sample Liver Segmentation Results. a** Comparison of lesion detection for all lesions; **b** Comparison of lesion detection for lesions smaller than 1 cm; **c** Comparison of lesion detection for lesions larger than or equal to 1 cm; **d** Examples of liver segmentation results in different image phases with various lesion types (green Line: contours of liver predicted by the AI system; red Line: contours of liver lesions predicted by the AI system; blue Line: contours of liver lesions labeled by doctors).'Source data are provided as a Source Data file (Source_data_Figure_6.xlsx).

Furthermore, a non-inferiority trial of 183 pathologically confirmed patients showed no significant difference between LiAIDS and radiologists in the binary classification (Accuracy: LiAIDS 0.962, radiologists 0.973, $p = 1.52$).For the seven-category classification, there were no significant differences between LiAIDS and radiologists for all types ($p > 0.5$) except ICC. For ICC, LiAIDS performed statistically better than radiologists (Accuracy: LiAIDS 0.879, radiologists 0.758, $p < 0.05$). These results demonstrated that LiAIDS has comparable or even better diagnostic performance than radiologists and can reduce the time and manual effort incurred by traditional diagnostic workflows. Figure 7b further depicted the distribution of lesion sizes and the distribution of lesion numbers for each type across all patients. In the non-inferiority study, the average lesion volume for patients with only benign lesions was $104.13 \pm 177.73$ cm$^3$, with the majority of lesions being HC ($46.68 \pm 234.69$ cm$^3$) and/or HH ($126.78 \pm 432.25$ cm$^3$). On the other hand, the mean lesion volume for patients with malignant lesions was $67.34 \pm 160.48$cm$^3$, with most lesions being HM ($15.38 \pm 77.70$ cm$^3$) or HCC ($112.23 \pm 226.68$ cm$^3$). The average number of lesions for patients with only benign lesions was $7.25 \pm 8.26$, while for patients with malignant lesions, it was $5.98 \pm 7.9$. The p-values for both lesion size and number were less than 0.001 between these two groups, indicating statistically significant differences. It is noteworthy that the 183 pathologically confirmed patients who took part in our non-inferiority study exceeded our initial sample size calculation for the trial. This estimation, based on a non-inferiority margin (delta, δ) of 0.1, a significance level (alpha, α) of 0.05, a power (1-beta, 1-β) of 0.8, and an expected efficacy accuracy of 0.92, had anticipated a need for 173 participants. Therefore, our study was adequately powered as the actual number of recruited participants surpassed the calculated necessity.

## Discussion

In this study, a fully automated liver lesion detection and diagnosis system known as LiAIDS was developed, which consisted of three deep learning models for lesion detection, liver segmentation, and lesion classification, respectively. The system was trained and validated using contrast-enhanced CT scans and clinical information from 12,610 patients enrolled in 18 hospitals. LiAIDS has demonstrated strong robustness and generalization ability both retrospectively and prospectively.

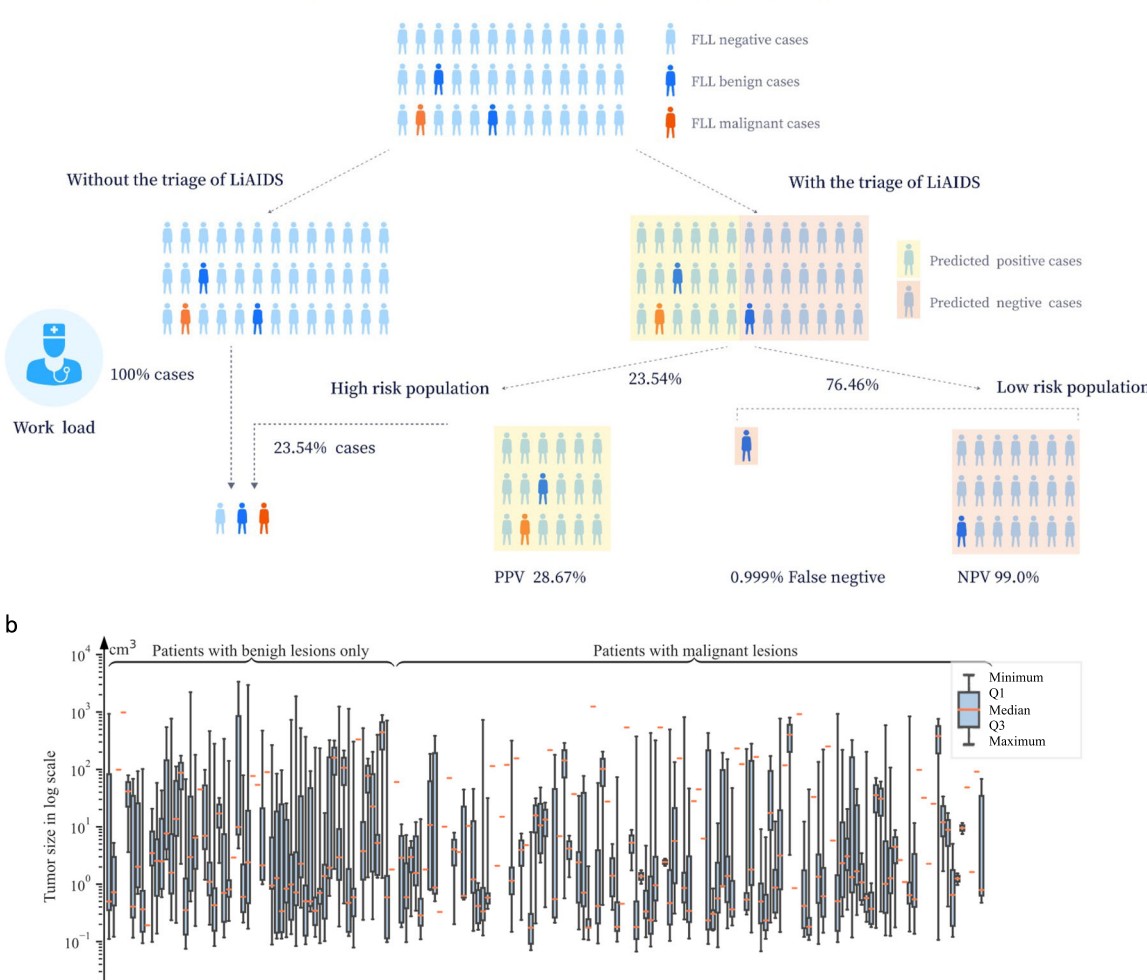

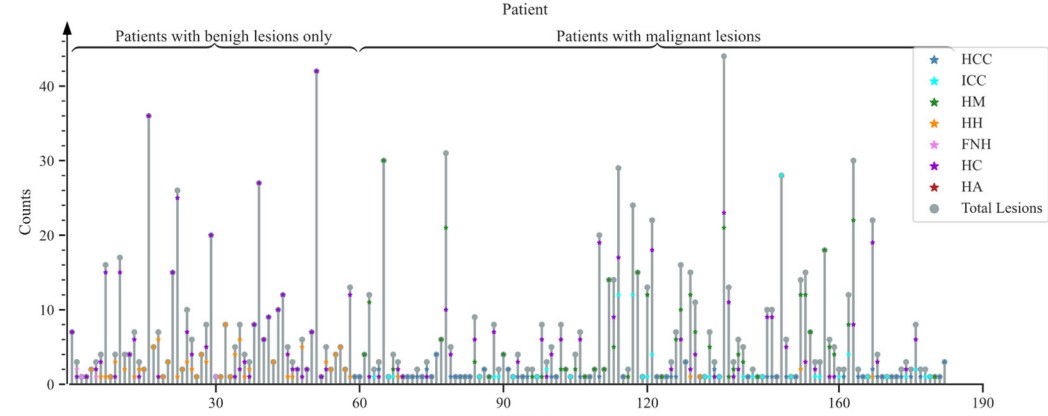

FLL=Focal Liver Lesions; LiAIDS=Liver Artificial Intelligence Diagnosis System; PPV=Positive Predictive Value; NPV=Negative Predictive Value; Q1=Lower Quartile; Q3=Upper Quartile; HCC=Hepatocellular Carcinoma; ICC=Intrahepatic Cholangiocarcinoma; HM=Hepatic Metastasis; HC=Hepatic Cyst; HH=Hepatic Haemangioma; FNH=Focal Nodular Hyperplasia; HA=Hepatic Abscess.

**Fig. 7 | Clinical Applications. a** Example of patient triage performed in Sir Run Run Shaw Hospital (SRRSH); **b** Distributions of lesion size and lesion numbers in each lesion type across all patients in patient triage. Source data are provided as a Source Data file (Source_data_Figure_7.xlsx).

Since the dataset used in this study was large enough and came from a relatively large number of hospitals, covering CT scanning devices from all major manufacturers worldwide (see Figure S3 and Table S7), LiAIDS is applicable to medical centers outside this study. Further experimental comparisons using single-center and multicenter training cohorts also demonstrated the superior performance of our large-scale multicenter study.

Compared with practicing radiologists, LiAIDS was superior to junior radiologists and comparable to senior radiologists. It is important to note that the medical hierarchy in China is structured into four

distinct levels: physician (equivalent to resident), attending physician, associate chief physician, and chief physician. In this study, we adopted the duration of practice in abdominal imaging diagnosis as our primary classification criterion. Specifically, junior radiologists are general radiologists with 5-10 years of experience in abdominal imaging diagnosis, which includes physicians and attending physicians. Conversely, senior radiologists are general radiologists with 10-20 years of experience, and this group comprises associate chief physicians and chief physicians.

To rigorously assess the performance of LiAIDS, we carried out an analysis at the patient level, recognizing that in clinical practice, the diagnosis of multiple FLLs in a single patient often depends on the most severe or clinically important lesion. As shown in Table 2, LiAIDS has demonstrated enhanced accuracy across all validation cohorts under the seven-category classification. Notably, accuracy improved from 0.880 (95% CI: 0.847-0.911) to 0.912 (95% CI: 0.879-0.941) in the internal cohort, from 0.872 (95% CI: 0.848-0.895) to 0.904 (95% CI: 0.878-0.928), 0.856 (95% CI: 0.831-0.881) to 0.886 (95% CI: 0.848-0.919), and 0.828 (95% CI: 0.798-0.859) to 0.884 (95% CI: 0.850-0.915) in the three external cohorts, and increased from 0.805 (95% CI: 0.777-0.833) to 0.833 (95% CI: 0.799-0.864) in the prospective cohort.

We further carried out a more thorough analysis on the distributions of CT acquisition parameters, particularly slice thickness and exposure mAs. Our findings indicated variations in accuracy rates across different ranges of slice thickness and exposure mAs. For instance, the accuracy rates of the seven-category classification in the internal validation cohort for different slice thicknesses were 0.729 (95% CI: 0.665-0.792), 0.892 (95% CI: 0.859-0.925), 0.835 (95% CI: 0.758-0.912) for ranges [2.5-5.0 mm], [5.0-7.0 mm), and [7.0-10.0 mm), respectively, all demonstrating significant differences (all $p < 0.001$). A similar trend was observed for exposure mAs, where the accuracy rates of the seven-category classification were 0.859 (95% CI: 0.809-0.905), 0.891 (95% CI: 0.842-0.939), 0.918 (95% CI: 0.852-0.984) for ranges [0-100], [100-200), and [200-500), respectively, in the internal validation cohort. We also investigated the effects of CT imaging parameters on the detection of lesions. The results, as shown in Table 6S, indicated less variations in recall rates across different slice thickness and exposure mAs ranges. For example, the recall rates for slice thickness ranges of [2.5-5.0 mm), [5.0-7.0 mm), and [7.0-10.0 mm) were 0.932, 0.940, and 0.907 respectively, with corresponding p-values of > 0.001, = 0.023, and <0.001 in the internal validation cohort. Similar trends were observed with regards to exposure mAs. More detailed findings and specific results can be found in Table 2 and Table S6.

Nevertheless, this study has several limitations. First, the sensitivity of ICC diagnosis remained low. To date, one previous article has reported the use of machine learning on CT images to diagnose ICC, with a reported sensitivity of 0.464[17]. In contrast, the sensitivity of LiAIDS for ICC diagnosis was 0.618 in the internal validation cohort, showing an improvement. Second, the training and external validation cohorts were retrospectively annotated, which may lead to some degree of bias. However, our prospective study suggested that this limitation may not be prominent. A third limitation is that this study was conducted at various centers in China only. Finally, the smooth integration of HIS (Hospital Information System), LIS (Laboratory Information System) and PACS (Picture Archiving and Communication System) systems in clinical workflows remain challenging in many medical centers.

In conclusion, a deep learning based AI system, LiAIDS, has been developed and validated both retrospectively and prospectively. This system can automated detect and differentiate FLLs using CECT scans and clinical information. We anticipate that LiAIDS will emerge as a valuable diagnostic aid, enhancing the efficiency and accuracy of radiologists via streamlining the diagnostic workflow, reducing patient waiting times, and augmenting the precision of diagnoses. Additionally, LiAIDS can also serve as an educational resource, offering real-time feedback and instructional support to radiology trainees. Moreover, in regions experiencing a scarcity of radiologists, LiAIDS can support the diagnostic process, promoting a more strategic allocation and effective use of medical human resources.

## Methods
### Ethical approval
This study has received approval from the Institutional Review Board (IRB) of Sir Run Run Shaw Hospital (SRRSH) and was carried out in adherence to the Declaration of Helsinki. Additionally, the prospective component of this study is officially registered with the Chinese Clinical Trial Registry, under the identifier ChiCTR2100045278 (accessible at [https://www.chictr.org.cn/showproj.html?proj=124700], registration date: April 10, 2021), and informed consent has been duly acquired. In parallel, all 17 collaborating institutions obtained requisite IRB approvals for their participation in the retrospective aspects of the study. Recognizing the non-invasive nature of the methodology and the anonymization of data, the IRB granted a waiver for the informed consent requirement.

### Data
This study included retrospective data from 11,385 patients managed in 18 hospitals in China between January 1st, 2010 and June 30th 2020, to develop and validate the proposed LiAIDS. In addition, a prospective study further included 1225 patients treated at SRRSH between July 1st, 2020 and June 30th, 2021. Furthermore, 13,192 consecutive patients admitted to SRRSH between May 1st 2022 and August 31th 2022 were collected for a study of patient triage. In this study, we laid out the following inclusion and exclusion criteria for data: Inclusion criteria were: (1) patients who underwent enhanced CT scans for FLLs and (2) patients aged 14 years and above. Exclusion criteria were: (1) patients who received any form of treatment for FLLs prior to the contrast-enhanced CT scan, including surgery, transcatheter arterial chemoembolization (TACE), radiofrequency ablation, chemotherapy, radiotherapy, targeted drug therapy, and immunotherapy; (2) patients who had a clinical diagnosis of malignant lesions but lacked pathological confirmation; (3) benign cases that lacked both a histopathological report and a consensus agreement; (4) cases with compromised CT image quality due to reasons including patient movement, incorrect positioning, presence of metallic objects, or equipment malfunctions; and (5) cases that lacked essential clinical information, including basic patient data (for example age and gender) and relevant medical history (such as hepatitis, cirrhosis, cholangiolithiasis, and extra-hepatic tumors). It needs to be noted that this study did not take into account chief complaints or lab findings. The design of this study took into account both sex and/or gender, with these attributes determined based on self-reporting at the time of patient enrollment. (See Fig. 2 for more details on data distribution and statistics). There were no adverse events in this study.

In this study, triple-phase CT scans were performed on all participants, including non-contrast phase, arterial phase, and portal venous phase. A non-contrast scan was performed before contrast injection, while the post-injection phases included the arterial phase (25-40 seconds after injection) and the portal venous phase (60-80 seconds after injection). CT scans were acquired using a slice collimation of 5/7 mm, a matrix of 512 ×512 pixels, and an in-plane resolution of 0.516-0.975 mm. The 18 hospitals in this study covered CT scanning equipment from all major global manufacturers, including Siemens, General Electric (GE), Philips, Toshiba, and United Imaging Healthcare (UIH). In our retrospective and prospective studies, each CT scan sequence was accompanied by a radiological report, which was initially generated by a radiologist after reading the scan and confirmed by a senior radiologist. In case of any disagreement, a final diagnostic decision was made at a departmental conference. (See Figure S3 and Table S7 for more details on data distribution and statistics).

### Radiologist annotation

Ten general radiologists with more than 5 years of experience in abdominal imaging diagnosis were divided into 5 groups of 2 to participate in data quality control and data annotation. Each FLL was manually annotated by a group of 2 general radiologists, with one radiologist delineating the boundaries of the lesion and/or liver under the supervision of another radiologist. The contours of the lesion and/or liver can be finalized when the two radiologists reached a consensus. The resulting lesion and liver delineation data were used as ground truth to train and validate LiAIDS lesion detection and liver segmentation models, respectively. The gold standard for lesion classification was established either from available histopathological reports or from the consensus of two senior general radiologists, each with over 20 years of experience in abdominal imaging diagnosis. Specifically, malignancies were validated via histopathology, while benign lesions were confirmed either through appropriate histopathology or by the joint agreement of the senior radiologists mentioned earlier. This agreement was achieved after an independent review of all pertinent information, which includes clinical data, CT scans, and associated radiological reports, collected over a follow-up period of at least six months. Cases that have neither a histopathological report nor a consensus agreement were all excluded from the study.

### LiAIDS algorithm development

The architecture of the proposed LiAIDS consisted of three main modules, namely lesion detection, liver segmentation, and lesion classification modules (Fig. 1). The lesion detection module aimed to identify and localize all potential FLL candidates on CT images. The use of liver segmentation was twofold. First, the segmented liver can serve as a false-positive identifier, filtering out lesions detected outside the liver region. Second, it can also be utilized as an atlas to correlate lesions independently detected in different CT phases. The lesion classification module was designed to use image features and clinical information to differentiate detected lesions into one of the seven most common types, and further classify them as malignant (HCC, ICC, HM) or benign (FNH, HH, HC, and HA). In this study, clinical information includes age, gender, medical history of hepatitis, history of liver cirrhosis, cholangiolithiasis, and extra-hepatic tumor and three different CT phases shared the same detection, segmentation, and feature extraction backbone models.

**Lesion Detection.** The network architecture of the lesion detection module was based on a two-stage detection algorithm called Faster R-CNN[21]. The first stage generated FLL candidates called proposals through a Region Proposal Network (RPN), while the second stage further classified FLL candidates and regressed their locations through a R-CNN network. CT is known to contain rich 3D contextual information and it is difficult to accurately detect lesions using features extracted from a single axial slice. As shown in Fig. 1a, we extended the state-of-the-art CSwin Transformer[20] from 2D to 3D, and applied the new 3D CSwin Transformer as the feature extraction backbone of the faster-RCNN framework. In this study, the input to our newly extended 3D CSwin Transformer backbone was composed of $2n + 1$ adjacent slices (i.e., the target slice and n adjacent slices directly above and below the target slice, in our case $n = 5$). To be able to detect lesions of various sizes, Feature Pyramid Network (FPN)[23] was also integrated into the RPN module. The p2 ~ p6 layers in the feature pyramid were used to generate lesion candidates through the parameter shared RPN network head.

**Liver Segmentation.** In this study, we specifically designed a hybrid convolutional network structure with 2D convolution as the main and 3D convolution as the auxiliary for liver segmentation. Such a design enabled the network to capture the 3D spatial information of the data while reducing memory usage. Furthermore, multiple attention

mechanisms including Non-Local attention[24] and scSE attention[25] were used to improve segmentation accuracy. As shown in Fig. 1b, our network adopted the encoder-decoder U-Net architecture[26], which was dominated by 2D convolutional blocks with 3D convolutional blocks located at the input, output and bottom layer of the network. Each 2D convolutional block consisted of two Conv2d units and an Efficient 2d Non-Local block[24]. The Conv2d unit splitted the unit's input into two even-numbered chunks channel-wisely and applied group convolutions with kernel size 5×5 and 7×7 to these two chunks, respectively. The output of the group convolution and the input of the unit were concatenated and fed to a scSE attention unit[25], followed by a sub-unit consisting of 1×1 Conv2d, BatchNorm (BN)[27], and GELU. Non-Local block[25] can help the network learn global features, thereby improving segmentation accuracy. Each 3D convolutional block consisted of 3x3x3 convolution, InstanceNorm[28], and Gaussian Error Linear Unit (GELU)[29]. 3D Non-Local blocks were added to the bottom layer of the network. Skip-connections between encoder and decoder were achieved by 1×1 convolutions and three deep supervision blocks[30] were applied to further improve segmentation performance. In our study, the loss function was cross entropy with Dice loss coefficient[31]. In our study, the average Dice coefficient (DICE) for 100 randomly selected cases from the external cohorts was 0.968, while for 100 randomly selected cases from the prospective cohort, it was 0.972. Examples of liver segmentation results were shown in Fig. 6b.

**Lesion Classification.** The lesion classification module included two parts: feature extraction based on image data and feature extraction based on clinical data. As shown in Fig. 1c, the image-based feature extraction network consisted of two sub-modules, the feature extraction backbone and the multi-phase feature-interaction. In this study, the feature extraction backbone was composed of three identical but independent networks for extracting image features of the arterial, venous, and non-contrast phases, respectively, where the multi-phase feature-interaction sub-module integrated fine-grained multi-phase features by capturing the subtle differences of lesion across different phases in a mutually reinforcing manner. More specifically, first, 3D CT image patches were extracted from the bounding box of each detected lesion and fed into the feature extraction backbone to obtain feature maps for all three CT phases independently, where DenseNet[32] combined with scSE attention unit[25] were used as the feature extraction backbone. Then, the feature maps extracted from three different CT phases were fed into the multi-phase feature-interaction sub-module via element-wise multiplication between feature map pairs. The newly generated feature maps were sequentially processed through adaptive pooling, concatenation, and two fully connected layers to obtain the final image features. Finally, the final image features and clinical features were concatenated and fed into two fully connected layers to output class probabilities. For each detected lesion, the output was a seven-dimensional vector representing the predicted probabilities for the seven disease categories. The category with the largest value in the seven-dimensional vector was taken as the final diagnosis of the lesion. In this study, the clinical information incorporated both numerical variables (such as age) and categorical variables (for example, gender and history of hepatitis). More detailed information can be found in the data inclusion and exclusion criteria in the Method section.

**Implementation details.** Lesion detection, liver segmentation and lesion classification models were independently trained. During the training of the lesion detection model, we applied transfer learning to pre-train the backbone network of the lesion detection module using the large-scale natural image dataset ImageNet[33]. The RGB channels of a natural image were treated as three consecutive slices. The model was trained using multi-scale input images of dimensions 384×384, 448×448, 512×512, 576×576, and 640×640. Online data augmentation

strategies were implemented, including horizontal and/or vertical flipping with a random probability ($p = 0.5$) and random rotations with a random probability ($p = 0.5$). Stochastic gradient descent was used with an initial learning rate of 0.01, a decay factor of 0.1 after the 8-th and 11-th epochs (12 epochs in total), and a weight decay of 0.0001. The batch size was set to 6. During the training of the liver segmentation model, online data augmentation strategies such as affine transformation, random cropping, and salt-and-pepper noise were also randomly applied. The adam optimizer[34] with an initial learning rate of 0.0005 decayed linearly with the number of iterations and a weight decay of 0.05, and a batch size of 4 were used. For training of the lesion classification model, the adam optimizer with an initial learning rate of 0.001, a decay factor of 0.1 for every 60 epochs, and a weight decay of 0.01 was used. The number of training epochs was set to 200, and the batch size was 16.

All codes were implemented in Python and Pytorch. Two workstations were used for individual model training and validation. More specifically, the lesion detection experiments were performed on a workstation platform with 4 NVIDIA RTX 2080 Ti GPUs with 11GB GPU memory, 256 G RAM, and Intel(R) Xeon(R) Gold6248 CPU @ 2.50 GHz, using Ubuntu 16.04. The liver segmentation and lesion classification experiments were performed on a workstation platform with 1 NVIDIA TITAN RTX GPU, 24GB GPU memory, 256 G RAM, and Intel(R) Xeon(R) Gold6248 CPU @ 2.50 GHz, using Ubuntu 16.04.

**Statistical analysis.** We employed Receiver Operating Characteristic (ROC) curves to evaluate our model's diagnostic performance. These curves were generated by varying the threshold for predicted probability and plotting the True Positive Rate (TPR, or sensitivity) against the False Positive Rate (FPR, 1-specificity). A high Area Under the Curve (AUC) indicates superior diagnostic capability. In our analysis, the AUC and confusion matrix were the primary metrics, providing comprehensive insights into model efficacy across different lesion types and addressing class imbalance. Additionally, we considered the following secondary metrics for performance evaluation: Accuracy, calculated as $(TP + TN)/(TP + TN + FP + FN)$, Sensitivity, defined as $(TP)/(TP + FN)$, Specificity, determined as $(TN)/(TN + FP)$, and Precision, computed as $(TP)/(TP + FP)$. Here, TP denotes True Positives, TN denotes True Negatives, FP denotes False Positives, and FN denotes False Negatives. For the comparison between AI and radiologists, the F1-score, the harmonic mean of sensitivity and precision was used as this metric offers a balanced view of both sensitivity and precision in a single measure. The F1-score is calculated as 2 * (sensitivity * precision) / (sensitivity + precision). All statistical analyses employed two-tailed tests, with p-values of 0.05 or lower deemed significant. These analyses were conducted using Python, version 3.7.6.

**Reporting summary**
Further information on research design is available in the Nature Portfolio Reporting Summary linked to this article.

## Data availability
All analytical data underpinning the findings of this study are incorporated within this paper in the designated Source Data files (Source_data_Figure_3.xlsx to Source_data_Figure_7.xlsx, Source_data_Figure_S1.xlsx to Source_data_Figure_S3.xlsx, Source_data_Table_1.xlsx to Source_data_Table_3.xlsx, and Source_data_Table_S1.xlsx to Source_data_Table_S7.xlsx).The original imaging and clinical datasets are subject to controlled access to prevent potential misuse, even though they have been anonymized. Access to the data is limited, as they were utilized under institutional permission, sanctioned by the institutional review board specifically for this study, and hence are not publicly accessible. For academic inquiries regarding the use of raw and processed data, please direct your requests via email to the corresponding author at henry_yinghn@zju.edu.cn. Each request will undergo an evaluation in accordance with institutional and departmental guidelines to ascertain if the requested data are bound by intellectual property rights or patient confidentiality commitments. The review process is expected to be completed within one month. Data sharing is exclusively for non-commercial academic purposes and will necessitate a formal material sharing agreement. Source data are provided with this paper.

## Code availability
The architecture of our system is an integration of innovative technologies, including an enhanced Faster R-CNN with the CSwin Transformer for feature extraction, a U-Net augmented with the scSE attention mechanism, and DenseNet. For transparency and reproducibility, the source code and models for each component are available through open-source platforms. The respective repositories are as follows: Faster R-CNN is hosted at https://github.com/open-mmlab/mmdetection; CSwin Transformer can be found at https://github.com/microsoft/CSWin-Transformer; U-Net is available at https://github.com/milesial/Pytorch-UNet; the scSE Attention Unit is located at https://gitcode.net/mirrors/shanglianlm0525/pytorch-networks/-/blob/master/Attention/SEvariants.py; and DenseNet can be accessed at https://github.com/xmuyzz/3D-CNN-PyTorch/blob/master/models/DenseNet.py. In addition, the full custom code is available in an open-source repository at https://github.com/DeepWiseAI/LiAIDS with detailed descriptions of the core functions in the form of pseudocode.

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

## Acknowledgements
This research was supported in part by National Key Research and Development Program of China (No.2021ZD0113302) [XQL, YML], the Natural Science Foundation of Zhejiang province, China (LQ20H160033) [HNY], the Fundamental Research Funds for the Central Universities (226-2022-00184) [HNY], and Hong Kong Research Grants Council under Collaborative Research Fund (Project No. HKU C7004-22G) [YZY].

## Author contributions
Xiujun Cai, Yizhou Yu, Wan Yee Lau, Xiaoqing Liu, Min Zhang, Shizheng Zhang, Hongjie Hu and Risheng Yu contributed to the study concept and design. Hanning Ying, Yiyue Ren, Shihui Zhen, Xiaojie Wang, Lian Duan Xiangdong Cheng, Xiangyang Gong, Haitao Jiang, Jianshuai Jiang, Jianjun Zheng, Kelei Zhu, Wei Zhou, Baochun Lu, Hongkun Zhou, Yiyu Shen, Jinlin Du, Mingliang Ying, Qiang Hong, Renya Jiang, Jingang Mo, Jianfeng Li, Guanxiong Ye and Mingzhi Cai contributed to acquisition of data. Bo Liu, Shizheng Zhang, Peng Hu, Hui Liu, Yiming Li, Xin Cheng, Xingxin Xu, Shuxin Wang and Huiping Bai contributed to analysis and interpretation of data. Hanning Ying, Xiaoqing Liu, Yizhou Yu, Min Zhang, Xiaoyin Xu, Long Jiao, Risheng Yu, Wan Yee Lau and Xiujun Cai contributed to writing, reviewing, and approval of the final version of this work.

## Competing interests
The authors declare no competing interests.

## Additional information

[1]Department of General Surgery, Sir Run Run Shaw Hospital, Zhejiang University School of Medicine, Hangzhou, China. [2]Deepwise Artificial Intelligence Laboratory, Beijing, China. [3]College of Computer Science and Technology, Zhejiang University, Hangzhou, China. [4]School of Medicine, Zhejiang University, Hangzhou, China. [5]Department of Radiology, Sir Run Run Shaw Hospital, Zhejiang University School of Medicine, Hangzhou, China. [6]Zhangzhou Municipal Hospital of Fujian Province, Zhangzhou, China. [7]Quzhou People's Hospital, Quzhou, China. [8]Cancer Hospital of the University of Chinese Academy of Sciences (ZheJiang Cancer Hospital), Hangzhou, China. [9]Zhejiang Provincial People's Hospital, Hangzhou, China. [10]Department of Hepatopancreatobiliary Surgery, Ningbo First Hospital, Ningbo, China. [11]Hwa Mei Hospital, University of Chinese Academy of Sciences (Ningbo No.2 Hospital), Ningbo, China. [12]Department of Hepatopancreatobiliary Surgery, Yinzhou People's Hospital, Ningbo, China. [13]Department of Radiology, Huzhou Central Hospital, Affiliated Central Hospital of Huzhou University, Huzhou, China. [14]Shaoxing People's Hospital, Shaoxing, China. [15]The First Hospital of Jiaxing Affiliated Hospital of Jiaxing University, Jiaxing, China. [16]The Second Hospital of Jiaxing Affiliated Hospital of Jiaxing University, Jiaxing, China. [17]Jinhua Municipal Central Hospital, Jinhua, China. [18]Jinhua GuangFU Hospital, Jinhua, China. [19]Taizhou Municipal Central Hospital, Taizhou, China. [20]The First People's Hospital of Wenling, Taizhou, China. [21]Lishui People's Hospital, Lishui, China. [22]Central Laboratory of Sir Run Run Shaw Hospital, Zhejiang University School of Medicine, Hangzhou, China. [23]Xiamen University, Xiamen, China. [24]Brigham and Women' Hospital, Harvard Medical School, Boston, MA, USA. [25]Faculty of Medicine, Imperial College London, London, UK. [26]Department of Radiology, Second Affiliated Hospital of Zhejiang University School of Medicine, Hangzhou, China. [27]Faculty of Medicine, the Chinese University of Hong Kong, Hong Kong, China. [28]Department of Computer Science, The University of Hong Kong, Hong Kong, China. [29]These authors contributed equally: Hanning Ying, Xiaoqing Liu, Min Zhang, Yiyue Ren, Shihui Zhen, Xiaojie Wang, Bo Liu. ✉e-mail: xxu@bwh.harvard.edu; l.jiao@imperial.ac.uk; risheng-yu@zju.edu.cn; josephlau@cuhk.edu.hk; yizhouy@acm.org; srrsh_cxj@zju.edu.cn

