## [Peer Review File · Nature Communications]

Reviewers' Comments:

Reviewer #1:

Remarks to the Author:

This study presents an AI system for automated detection and diagnosis of focal liver lesions based on CT image and clinical data. Strengths of the study include: (1) development of end-to-end deep learning models for automated detection and diagnosis of focal liver lesions; (2) multi-center validation with very large patient cohorts; (3) prospective evaluation of the deep learning model. The system may be used as a first read to make more efficient diagnosis and reduce radiologist workload. It could potentially improve diagnostic accuracy in low-resource settings with insufficient radiology expertise.

My specific comments are below.

1. The number of lesions for each of the 7 lesion classes should be shown for each dataset. Is there a sufficient number of lesions within each class to assess the model, especially in the prospective study?
2. Figure 3a-e and where applicable: AUC for binary classification (one vs. others) is an overly optimistic measure to evaluate model performance in multi-class problems. The confusion matrix showing absolute numbers should be presented to comprehensively evaluate the accuracy of multi-class diagnosis.
3. Figure 3f: There is significant overlap between ICC and other malignant lesions in the t-SNE map. There is also a good amount of overlap between two types of benign lesions FNH and HH. This is concerning. Any comments or plan to address this issue?
4. Figure 3g-h: How do the probability scores reflect model performance?
5. Figure 4: study is designed for evaluating radiologist against AI performance. Any thoughts on how AI may improve upon radiologist performance through human-AI collaboration?
6. Figure 6: lesion detection. About 5-10% small lesions are missed by AI. What are the characteristics of these missed lesions, benign or malignant? Do they significantly overlap with those missed by radiologists?
7. What are the distributions of CT acquisition parameters such as slice thickness, mAs levels, etc? Do they affect lesion detection or diagnosis by the AI model?
8. What's the accuracy of liver segmentation?

Reviewer #2:

Remarks to the Author:

This paper presents a large, multi-center study for evaluation of a machine learning model for lesion classification on CT images. The results show that the proposed method is promising for clinical applications.

Figures:

Figure 3: One of the legends state "Marco average" for Figures 3A- 3E. I believe it should be "Macro Average"

Figure 4: Having titles for each subplot would be helpful to understand the information on the graphs. In addition, Figure 4D does not have a label for the Y axis, which should be added.

Results:

You mentioned that your method for the deployment in triage applications identified 76.46% as low risk patients, with an NPV of 99.0%. Please provide the NPV of the current clinical practice to understand how your method compares to the current standard of care. In addition, please provide the positive predictive value (PPV), or the false positive rate for the triage application, along with the current standard of care comparison. If the PPV decreases with the usage of the device, then the patients may need to be falsely recalled more often, putting additional burden, thus both the NPV and the PPV should be considered for the performance of your method.

When reporting the results in the text, I believe reporting the performance results on the external validation cohorts is more impactful as it shows independent testing and that the proposed method performs well at completely new sites.

On page 13, you mention that "the performance of LiAIDS on the basis of CECT images and clinical information significantly improved the accuracy of FLL classification." However, there was no statistical testing to show that the improvement is needed significant. The p-values were also not mentioned in Figure 5. Please perform the statistical testing and show the p-values. If the difference is not statistically significant, then please correct the text to reflect the results.

On page 13, you mentioned that "[m]ore importantly, there were statistically significant differences in accuracy, specificity and precision between models using multicenter and single-center training data." This is not true for all sites (specifically, the precision for QZH, NBH are not significant). The text implies that all the differences in the mentioned metrics were statistically significantly different. Please update the text to accurately reflect the results.

On Figure 3C, the results for HA AUC for site QZH is much lower compared to all other sites and results. Please provide an explanation why the results are so low. In addition, the lower confidence interval for the HA AUC is 0.0, which seems incorrect given the results.

You had separated out 3 sites for external validation. Were there any methods used to determine which sites were set aside to be the external test cohorts?

On page 14, you mention that LiAIDS required an average of 128 seconds to report a diagnosis after receiving CT images. However, in your methods section, you mentioned that the lesion detection and lesion segmentation/classification was performed on two separate workstations. It is unclear how you calculated the total time to report a diagnosis when the experiments were conducted on two separate hardware. Please clarify.

Methods:

Did the IRBs of the 18 sites approve the retrospective study at their institution, or was informed consent waived?

Page 17: For the faster R-CNN lesion Detection implementation, was any part of the faster R-CNN changed from the cited paper, such as the NMS parameter, or were they left as original?

A table with the number of patients with each of the 7 categories should be provided to understand the sample size of the each of the 7 conditions.

Reviewer #3:

Remarks to the Author:

In this study, authors developed LiAIDS which detect focal liver lesions (FLLs) and categorize them into either benign/malignant and assign them into common liver diseases (HCC, iCCA, mets, cyst, adenoma and FNH) from 11,385 patients CT data from 18 hospitals in China. LiRADS were included three modules (liver segmentation, FLL detection and diagnosis), and each of them were tested in the internal validation cohort and external validation cohort. Authors found that LiAIDS had high diagnostic performance of assigned categorization of FLLs, binary classification. And LiAIDS outperformed junior radiologists for binary classification and outperformed all regarding sensitivity and specificity. Patients triage was also successful with NPV of 99%, and the reporting time was less than 5 min which was significantly shorter than radiologists' reporting time.

Strengths: a) a large number of training and testing data; b) good performance of the model for clinical practice; c) a clinically relevant topic

Weakness: a) uncertain/unclear methodology; b) unclear clinical context

General comments

The use of AI is one of the hottest topics in Radiology, and vigorous attempts have been made to

implement the AI in our practice. The scale of training data and the thorough tests of the LiAIDS are impressive. However, I am afraid that I have concerns about the methodology and whether the superiority of the model's performance is a bit overrated. In addition, this article includes too many facets of the study from model accuracy evaluation in aspects of techniques to clinical implications, which may hamper readability.

Title: ok

Abstract: please provide data in the abstract rather than describing simple introduction with phrases. And what does 'systematic study' mean in this context?

Key words: consider including triage and diagnosis, instead of clinical data

Introduction

Ok. But it seems to be too long. Some of the contents can be moved to discussion or removed.

Methods

First, please clarify the eligibility criteria in the main body. Mentioning it in only figure is not acceptable. And in figure 2, 'any treatment' means both systemic and locoregional therapy? And indeed, 7 mm slice thickness (p.16) is also not regarded as appropriate image quality.

In addition, the reference standard is semi-solid; malignancies were confirmed histologically, but the benign lesions were categorized by two radiologists' consensus and follow-up data. Therefore, there must be some 'indeterminate' FLLs in which follow-up data were not available, too-small-to-characterize, or non-specific finding to be assigned to any of seven FLL categories. Were those patients excluded? In addition, the definition of 'clinical information' is not clear- does this refer to patients' chief complaint only or underlying disease or lab findings or all of them? (please don't be bothered if you mention the details in Table S7, since Table S7 is not available for me of unknown reasons.)

And there is no information regarding the number of FLLs per patient. The size of FLLs needs to be mentioned as well. Regarding the statistical analysis, if multiple FLLs exist in a patient, how did authors deal with that?

Regarding the clinical application in patient triage, I wonder how the reading time was recorded in radiologists since it is too long.

Please provide the criteria of 'correct detection' by LiAIDS: any criteria of dice score or others regarding the FLL segmentation?

How was the sample size determined in non-inferiority trial of 183 pathologically confirmed high risk patients (p. 14)?

Results

As I commented earlier, the reading time of radiologists is unusually long. Does this the 'review time' for this study? In other words, the time to record annotate all FLLs? If so, it is not a reading time to generate report. In addition, if it is a measurement of radiologists to review CT scans and then generate report, the comparison between the radiologists and LiAIDS is not fair because radiologists should mention extrahepatic findings.

Also, per-lesion accuracy does not often reflect clinical relevance. Multiple hepatic cysts does not matter and detecting/reporting all hepatic cysts are not recommended since it is clinically irrelevant. In addition, patients with multiple liver metastases, all lesion detection may not be relevant either for the same reason.

And please clarify the definition of junior and senior radiologists. Radiologists implies that they are board certified, and '5 to 10 years' experience of radiology' should mean that they have 5 to 10 years' experience in liver imaging (reading the films alone for clinical purpose, not research purpose) after having radiology board. Do authors mean this, and 'junior radiologist' does not indicate radiology resident or fellows? I request authors clarify this and whether they are body radiologist or general radiologist as well since your description in Discussion.

In addition, the results of non-inferiority study cannot be easily interpreted without number of each FLLs and the size of the tumor in addition to numbers of FLLs per patient.

Discussion

Authors claimed that junior radiologists review the image initially and senior radiologists confirm it. I do not know much about Chinese radiology training system, but in most countries, this is only for radiology residents or fellows. No board-certified radiologists work in this manner. So it cannot be generalized and should be removed.

Response Letter

6/21/2023

Dear Reviewer #1, Reviewer#2, and Reviewer #3,

We would like to express our sincere appreciation for your diligent efforts and constructive feedback on our manuscript entitled "A Multicenter Study of a Clinically Applicable AI System for Automated Detection and Diagnosis of Focal Liver Lesions" (NCOMMS-23-00871). Your invaluable comments have greatly contributed to improving the quality of our work.

Please find enclosed the revised manuscript with tracked changes. In the following section, we provide detailed responses addressing each of your comments.

REVIEWER COMMENTS

Reviewer #1 (Remarks to the Author):

This study presents an AI system for automated detection and diagnosis of focal liver lesions based on CT image and clinical data. Strengths of the study include: (1) development of end-to-end deep learning models for automated detection and diagnosis of focal liver lesions; (2) multi-center validation with very large patient cohorts; (3) prospective evaluation of the deep learning model. The system may be used as a first read to make more efficient diagnosis and reduce radiologist workload. It could potentially improve diagnostic accuracy in low-resource settings with insufficient radiology expertise.

Response: We genuinely appreciate your positive and encouraging comments and endorsements above. Additionally, we are grateful for your constructive and insightful suggestions provided below.

My specific comments are below.

1. The number of lesions for each of the 7 lesion classes should be shown for each dataset. Is there a sufficient number of lesions within each class to assess the model, especially in the prospective study?

Response: We appreciate your insightful observation and the opportunity to provide further clarity on this issue. As per your suggestion, we have elaborated on the distribution of lesions among each of the seven lesion classes across all datasets. As a common rule of thumb in

traditional statistical studies, it's usually recommended to have at least 30 instances for each class in the test set [1]. Although somewhat arbitrary, this guideline typically strikes a decent balance between achieving statistical significance and managing computational expense. In the context of our prospective study, the counts for each of the 7 lesion classes all exceed this threshold, as detailed in the revised Table 1. Consequently, we are confident that the number of lesions in each class is ample for reliable model assessment.

[1] Whitehead AL, Julious SA, Cooper CL, Campbell MJ. Estimating the sample size for a pilot randomised trial to minimise the overall trial sample size for the external pilot and main trial for a continuous outcome variable. *Stat Methods Med Res.* 2016 Jun;25(3):1057-73. doi: 10.1177/0962280215588241. Epub 2015 Jun 19. PMID: 26092476; PMCID: PMC4876429.

Location of changes in the revised manuscript (marked in blue): Please refer to the revised Table 1 and the text on page 4, lines 184-185 for the specific modifications.

2. Figure 3a-e and where applicable: AUC for binary classification (one vs. others) is an overly optimistic measure to evaluate model performance in multi-class problems. The confusion matrix showing absolute numbers should be presented to comprehensively evaluate the accuracy of multi-class diagnosis.

Response: We are grateful for your insightful and constructive suggestion. In response, we have included confusion matrices in the revised manuscript, as you recommended.

Location of changes in the revised manuscript (marked in blue): Please refer to the updated Figure 3, the text on page 5, lines 211-214, and lines 234-237 for the specific modifications.

3. Figure 3f: There is significant overlap between ICC and other malignant lesions in the t-SNE map. There is also a good amount of overlap between two types of benign lesions FNH and HH. This is concerning. Any comments or plan to address this issue?

Response: We appreciate your insightful observation and the opportunity to provide further clarity on this issue. As depicted in the updated Figure 3f, the intersection between ICC and other malignant lesions on the t-SNE map appeared less prominent as we expanded the lesion sample size from 379 to 437. Additionally, we undertook an in-depth analysis of those FNH and HH samples that exhibited intertwined characteristics. Remarkably, both categories demonstrated significantly smaller mean volumes (cm^3) compared to the overall volume distribution within their respective data cohort (FNH: 6.309 ± 5.72 vs 34.8 ± 139.5 and HH: 2.457 ± 2.36 vs 19.8 ± 46.8). In a clinical setting, diagnosing of FNH and/or HH can be challenging, particularly in the case of smaller lesions. The usual procedure for managing such cases typically involves additional imaging studies, predominantly MRI, or ongoing clinical assessments. Given the benign nature of these conditions, immediate differentiation is not vital.

However, diligent and continued monitoring of these conditions is essential due to potential alterations and progression over time.

Location of changes in the revised manuscript (marked in blue): Please refer to the updated Figure 3 and the text on page 5, lines 211-224 for the specific clarifications.

4. Figure 3g-h: How do the probability scores reflect model performance?

Response: We appreciate your attention to detail. In Figure 3g-h, the probability scores represent the prediction confidence level of the model. Additionally, the distribution of probability scores among different cohorts provides insights into the generalizability of the model. A consistent distribution across cohorts indicates a high level of performance consistency, suggesting strong generalizability.

Location of changes in the revised manuscript (marked in blue): Please refer to the text on page 5, lines 229-233 for further clarifications.

5. Figure 4: study is designed for evaluating radiologist against AI performance. Any thoughts on how AI may improve upon radiologist performance through human-AI collaboration?

Response: We appreciate your feedback. In response to your insightful and constructive comments, we have conducted additional experiments on human-AI collaboration. These experiments have been included in the revised manuscript. We believe that these additions enhance the comprehensiveness and relevance of our study.

Location of changes in the revised manuscript (marked in blue): Please refer to the updated Figure 4, updated Table 3, updated Table S2, and the text on pages 5-6, lines 249-288 for the specific modifications.

6. Figure 6: lesion detection. About 5-10% small lesions are missed by AI. What are the characteristics of these missed lesions, benign or malignant? Do they significantly overlap with those missed by radiologists?

Response: Thank you for bringing this to our attention. We have carefully considered your insightful and constructive comments and have conducted further analysis on missed lesions. The results of this analysis have been included in the revised manuscript. We believe that these additions contribute to a more comprehensive understanding of our study findings.

Location of changes in the revised manuscript (marked in blue): Please refer to the text on page 8, lines 393-396 for modifications.

7. What are the distributions of CT acquisition parameters such as slice thickness, mAs levels, etc? Do they affect lesion detection or diagnosis by the AI model?

Response: *We appreciate your valuable input. Taking into account your insightful and constructive comments, we have conducted further analysis on the distributions of CT acquisition parameters, specifically focusing on slice thickness and mAs levels. We have included the results of this analysis in the revised manuscript. These additions provide a more comprehensive understanding of the impact of acquisition parameters on our study.*

Location of changes in the revised manuscript (marked in blue): *Please refer to the updated Table 2, updated Table S6, and the text on pages 9-10, lines 481-498 for modifications.*

8. What's the accuracy of liver segmentation?

Response: *We appreciate your observation. In response to your feedback, we have included the accuracy of liver segmentation in our revised manuscript. This addition provides important information regarding the performance of the segmentation component of our study. We thank you for bringing this to our attention and for contributing to the improvement of our manuscript.*

Location of changes in the revised manuscript (marked in blue): *Please refer to the text on page 12, lines 627-629 for the specific modifications.*

Reviewer #2 (Remarks to the Author):

This paper presents a large, multi-center study for evaluation of a machine learning model for lesion classification on CT images. The results show that the proposed method is promising for clinical applications.

Response: *We sincerely appreciate your positive and encouraging comments and endorsements expressed above. Furthermore, we are grateful for your constructive and insightful suggestions provided below. Your feedback has been instrumental in enhancing the quality and clarity of our work. Thank you for your valuable contribution to our manuscript.*

Figures:

Figure 3: One of the legends state "Marco average" for Figures 3A- 3E. I believe it should be "Macro Average"

Response: *Thank you for bringing this to our attention. We have made the corresponding corrections in our revised manuscript. We appreciate your attention to detail and your contribution to improving the accuracy of our work.*

Location of changes in the revised manuscript (marked in blue): Please refer to the updated Figure 3A-E for the specific corrections.

Figure 4: Having titles for each subplot would be helpful to understand the information on the graphs. In addition, Figure 4D does not have a label for the Y axis, which should be added.

Response: Thank you for bringing this to our attention. We have made the corresponding corrections in our revised manuscript. We appreciate your attention to detail and your contribution to improving the accuracy of our work.

Location of changes in the revised manuscript (marked in blue): Please refer to the updated Figure 4D for the specific corrections.

Results:

You mentioned that your method for the deployment in triage applications identified 76.46% as low risk patients, with an NPV of 99.0%. Please provide the NPV of the current clinical practice to understand how your method compares to the current standard of care. In addition, please provide the positive predictive value (PPV), or the false positive rate for the triage application, along with the current standard of care comparison. If the PPV decreases with the usage of the device, then the patients may need to be falsely recalled more often, putting additional burden, thus both the NPV and the PPV should be considered for the performance of your method.

Response: We appreciate your insightful suggestion. In response, we have included the positive predictive value (PPV) in the revised manuscript. Thank you for your valuable input and for helping us improve the manuscript. While there is no specific report on the current standard of care, a recent study on 214 patients [22] that concentrated on identifying malignancy in FLLs found that standard ultrasound had an NPV of 78% and a PPV of 60%. In contrast, CT scans yielded an NPV of 86% and a PPV of 81%, and MRI provided an NPV of 91% and a PPV of 68%. We must underscore that our goal is to reduce radiologists' workload by minimizing the number of cases they need to review, not to replace them with the LiAIDS system for diagnostic purposes. As such, a lower PPV is not a major concern in this context. The triage process using LiAIDS can decrease radiologists' workload by 76.46%, while resulting in less than 1% false negative cases.

Location of changes in the revised manuscript (marked in blue): Please refer to the updated Figure 7 and the text on page 8, lines 410-421 for the specific modifications.

When reporting the results in the text, I believe reporting the performance results on the external validation cohorts is more impactful as it shows independent testing and that the proposed

method performs well at completely new sites.

Response: *We are grateful for your insightful suggestion. In accordance with your recommendation, we have updated the revised manuscript by including additional performance results on the external validation cohorts. These additions provide a more comprehensive evaluation of the model's performance. Thank you for your valuable input, which has contributed to the enhancement of our manuscript.*

Location of changes in the revised manuscript (marked in blue): *Please refer to the updated Figure 5 and the updated Table S3 for the specific modifications. Consequently, all performance results have been reported for the external validation cohorts or all cohorts, with the exception of the AI vs radiologists comparison, which was conducted exclusively on the prospective cohort.*

On page 13, you mention that “the performance of LiAIDS on the basis of CECT images and clinical information significantly improved the accuracy of FLL classification.” However, there was no statistical testing to show that the improvement is needed significant. The p-values were also not mentioned in Figure 5. Please perform the statistical testing and show the p-values. If the difference is not statistically significant, then please correct the text to reflect the results.

Response: *We appreciate your insightful and constructive suggestion. In response, we have conducted the corresponding statistical testing and included the p-value in Figure 5 of the revised manuscript. Additionally, we have modified the text accordingly to align with your suggestion. These revisions strengthen the statistical analysis and improve the clarity of our findings. Thank you for your valuable input, which has positively contributed to the overall quality of our manuscript.*

Location of changes in the revised manuscript (marked in blue): *Please refer to the updated Figure 5, updated Table S3, and the text on pages 6-7, lines 291-324 for modifications.*

On page 13, you mentioned that “[m]ore importantly, there were statistically significant differences in accuracy, specificity and precision between models using multicenter and single-center training data.” This is not true for all sites (specifically, the precision for QZH, NBH are not significant). The text implies that all the differences in the mentioned metrics were statistically significantly different. Please update the text to accurately reflect the results.

Response: *Thank you for your feedback. We have carefully revised the manuscript accordingly. We appreciate your attention to detail and your contribution to improving the overall quality of our work.*

Location of changes in the revised manuscript (marked in blue): *Please refer to the text on page 7, lines 331-346 for the specific modifications.*

On Figure 3C, the results for HA AUC for site QZH is much lower compared to all other sites and results. Please provide an explanation why the results are so low. In addition, the lower confidence interval for the HA AUC is 0.0, which seems incorrect given the results.

Response: *We appreciate your observation. The reason for the significantly lower HA AUC results for site QZH compared to all other sites is due to the small number of HA cases in QZH, as indicated in the updated Table 1 in the revised manuscript. Specifically, there were only 2 lesions in QZH that were not correctly identified, resulting in a lower confidence interval for the HA AUC of 0.0. We have provided these clarifications in the revised manuscript to address this issue. Thank you for bringing this to our attention and for your contribution to the accuracy and transparency of our findings.*

Location of changes in the revised manuscript (marked in blue): *Please refer to the updated Table 1, Figure 3 (j), and the text on page 5, lines 235-237 for further clarifications.*

You had separated out 3 sites for external validation. Were there any methods used to determine which sites were set aside to be the external test cohorts?

Response: *We appreciate your feedback. For the purpose of external validation, we specifically selected the three largest sites, excluding the largest SRRSH, to form our external validation cohorts. This approach guarantees a comprehensive performance assessment over a wide range of data, thereby capturing a greater diversity of patient scenarios. We sincerely thank you for raising this concern, and highly value your contribution to enhancing the transparency of our study.*

Location of changes in the revised manuscript (marked in blue): *Please refer to the text on page 4, lines 161-163, and lines 163-167, for further clarifications.*

On page 14, you mention that LiAIDS required an average of 128 seconds to report a diagnosis after receiving CT images. However, in your methods section, you mentioned that the lesion detection and lesion segmentation/classification was performed on two separate workstations. It is unclear how you calculated the total time to report a diagnosis when the experiments were conducted on two separate hardware. Please clarify.

Response: *Thank you for your feedback. We appreciate the opportunity to clarify the distinction between the system inference phase and individual model training/validation phase. The reported average of 128 seconds pertains specifically to the overall system testing/inference phase, where the lesion detection, segmentation, and classification were performed on a single workstation. On the other hand, the mention of lesion detection and lesion segmentation/classification being performed on separate workstations in the methods section*

refers to the individual model training/validation phase. We apologize for any confusion caused by the initial lack of clarity and appreciate your understanding. The revised manuscript has been updated to include this clarification, ensuring a clearer understanding of our study methodology. Thank you for bringing this to our attention, and we value your contribution to the accuracy and transparency of our work.

Location of changes in the revised manuscript (marked in blue): Please refer to the text on page 8, lines 401-403 and page 13, lines 674-675 for further clarifications.

Methods:

Did the IRBs of the 18 sites approve the retrospective study at their institution, or was informed consent waived?

Response: Thank you for your comment. This study has received approval from the IRB of SRRSH and was carried out in adherence to the Declaration of Helsinki. The prospective aspect of this study has been registered in the Chinese Clinical Trial Registry (Registration No. ChiCTR2100045278). Furthermore, each of the other 17 participating sites received approval from their respective IRB for their involvement in this retrospective study, with the requirement for informed consent being waived.

Location of changes in the revised manuscript (marked in blue): Please refer to the text on page 10 lines 518-523 for this specific clarifications.

Page 17: For the faster R-CNN lesion Detection implementation, was any part of the faster R-CNN changed from the cited paper, such as the NMS parameter, or were they left as original?

Response: Thank you for your comment. We indeed maintained the same parameters as mentioned in the cited paper. We greatly appreciate your keen attention to detail and your valuable contribution towards enhancing the transparency of our study.

Location of changes in the revised manuscript (marked in blue): Please refer to the text on page 7, line 368-369 for further clarifications.

A table with the number of patients with each of the 7 categories should be provided to understand the sample size of the each of the 7 conditions.

Response: Thank you for bringing this to our attention. We wholeheartedly agree with the importance of providing detailed information regarding the number of patients in each of the 7 categories for all datasets. In response to your feedback, we have included these details in the revised manuscript. This addition enhances the transparency and comprehensiveness of our study, enabling readers to gain a clearer understanding of the patient distribution across

different categories. We greatly appreciate your valuable input, which has greatly contributed to the enhancement of our manuscript.

Location of changes in the revised manuscript (marked in blue): Please refer to the updated Table 1 for the revised modifications.

Reviewer #3 (Remarks to the Author):

In this study, authors developed LiAIDS which detect focal liver lesions (FLLs) and categorize them into either benign/malignant and assign them into common liver diseases (HCC, iCCA, mets, cyst, adenoma and FNH) from 11,385 patients CT data from 18 hospitals in China. LiRADS were included three modules (liver segmentation, FLL detection and diagnosis), and each of them were tested in the internal validation cohort and external validation cohort. Authors found that LiAIDS had high diagnostic performance of assigned categorization of FLLs, binary classification. And LiAIDS outperformed junior radiologists for binary classification and outperformed all regarding sensitivity and specificity. Patients triage was also successful with NPV of 99%, and the reporting time was less than 5 min which was significantly shorter than radiologists' reporting time.

Strengths: a) a large number of training and testing data; b) good performance of the model for clinical practice; c) a clinically relevant topic

Response: *We sincerely appreciate your positive and encouraging comments and endorsements expressed above. Furthermore, we are grateful for your constructive and insightful suggestions provided below.*

Weakness: a) uncertain/unclear methodology; b) unclear clinical context

Response: *Thank you for your feedback highlighting the weaknesses related to the methodology and clinical context in our manuscript. We acknowledge these concerns and have taken significant steps to address them in the revised version. We have provided more clarity regarding the methodology, ensuring that it is well-described and understandable. Additionally, we have included additional details and explanations to enhance the understanding of the clinical context in which our study is situated. We appreciate your valuable input, which has helped us improve the quality and readability of our work. Thank you for your contribution and support throughout the review process.*

General comments

The use of AI is one of the hottest topics in Radiology, and vigorous attempts have been made

to implement the AI in our practice. The scale of training data and the thorough tests of the LiAIDS are impressive. However, I am afraid that I have concerns about the methodology and whether the superiority of the model's performance is a bit overrated. In addition, this article includes too many facets of the study from model accuracy evaluation in aspects of techniques to clinical implications, which may hamper readability.

Response: Thank you for your feedback and positive comments regarding the use of AI in Radiology and the scale of training data and tests conducted. We appreciate your concerns about the methodology and the potential overrating of the model's performance. In response to your suggestions, we have made substantial revisions to the manuscript to address these concerns and enhance readability. We value your input and appreciate your contribution to the improvement of our manuscript.

Title: ok

Response: Thank you for your feedback.

Abstract: please provide data in the abstract rather than describing simple introduction with phrases. And what does 'systematic study' mean in this context?

Response: Thank you for your insightful suggestion. We have added more details about the data used in the abstract to provide a clearer understanding of the study. Additionally, we removed the term "systematic" to avoid any confusion. We appreciate your valuable input and your contribution to enhancing the quality of our work.

Location of changes in the revised manuscript (marked in blue): Please refer to the revised Abstract for modifications.

Keywords: consider including triage and diagnosis, instead of clinical data

Response: Thank you for your insightful suggestion. We have revised the Keywords as suggested.

Location of changes in the revised manuscript (marked in blue): Please refer to Keywords for modifications.

Introduction

Ok. But it seems to be too long. Some of the contents can be moved to discussion or removed.

Response: Thank you for your insightful suggestion. We have worked on making the introduction more concise by removing unnecessary details. We appreciate your valuable input, which has contributed to improving the structure and readability of our work. Thank you for your contribution and support throughout the review process.

Methods

First, please clarify the eligibility criteria in the main body. Mentioning it in only figure is not acceptable. And in figure 2, ‘any treatment’ means both systemic and locoregional therapy? And indeed, 7 mm slice thickness (p.16) is also not regarded as appropriate image quality.

Response: Thank you for your insightful feedback. In response, we've added further clarification to the eligibility criteria in the study's main body and elaborated the definition of "any treatment" to encompass surgery, transcatheter arterial chemoembolization (TACE), radiofrequency ablation, chemotherapy, radiotherapy, targeted drug therapy, and immunotherapy. We've also carried out a more thorough analysis on the distributions of CT acquisition parameters, particularly slice thickness. As indicated in the updated Table 2 and Table S6, most of the data centers have data with a slice thickness of less than 7mm, with the exception of the internal and prospective data cohorts, which contain data with a slice thickness of 7mm and above. Nonetheless, to bolster the robustness and adaptability of the system, we included images with a slice thickness of 7mm and above in our study.

Location of changes in the revised manuscript (marked in blue): Please refer to the text on pages 10-11, lines 531-545, the updated Table 2, and the updated Table S6 for additional clarifications.

In addition, the reference standard is semi-solid; malignancies were confirmed histologically, but the benign lesions were categorized by two radiologists’ consensus and follow-up data. Therefore, there must be some ‘indeterminate’ FLLs in which follow-up data were not available, too-small-to-characterize, or non-specific finding to be assigned to any of seven FLL categories. Were those patients excluded?

Response: Thank you for your thoughtful feedback. The gold standard for lesion classification was established either from available histopathological reports or from the consensus of two senior general radiologists, each with over 20 years of experience in abdominal imaging diagnosis. Specifically, malignancies were confirmed through histopathology, while benign lesions were authenticated either through appropriate histopathology or by the collective agreement of the aforementioned senior radiologists. This consensus was reached after independently reviewing all pertinent information, including clinical data, CT scans, and associated radiological reports, collected over a follow-up period of at least six months. As outlined in the section detailing data inclusion and exclusion criteria, benign cases that have

neither a histopathological report nor a consensus agreement were all excluded in this study. As a result, patients presenting with 'indeterminate' FLLs for which follow-up data was unavailable, lesions that were too small to characterize, or findings too non-specific to be assigned to any of the seven FLL categories, were excluded in this study.

Location of changes in the revised manuscript (marked in blue): Please refer to the text on page 11, lines 568-577 for the specific clarifications.

In addition, the definition of 'clinical information' is not clear- does this refer to patients' chief complaint only or underlying disease or lab findings or all of them? (please don't be bothered if you mention the details in Table S7, since Table S7 is not available for me of unknown reasons.)

Response: We appreciate your keen observation. Our study incorporated basic patient information (such as age and gender) along with pertinent medical history encompassing hepatitis, cirrhosis, cholangiolithiasis, and extra-hepatic tumor. This compilation of clinical information facilitated a broader evaluation of our subjects and offered vital context when assessing our AI system's performance. However, it needs to be noted that this study did not take into account chief complaints or lab findings.

Location of changes in the revised manuscript (marked in blue): Please refer to the text on page 6, lines 292-295, and on page 10, lines 541-544 for additional clarifications.

And there is no information regarding the number of FLLs per patient. The size of FLLs needs to be mentioned as well. Regarding the statistical analysis, if multiple FLLs exist in a patient, how did authors deal with that?

Response: Thank you for bringing this to our attention. We have provided more details regarding the number of focal liver lesions (FLLs) per patient and the size of FLLs. More importantly, we have included results on patient-level analysis as we agree that a patient-level analysis would be valuable to further enhance the understanding of FLL diagnosis. In clinical practice, when multiple FLLs are present in a patient, the diagnosis is typically based on the most severe or clinically relevant lesion. We believe that this addition will provide a more comprehensive understanding of the study findings and enhance the overall quality of the manuscript. We thank you for your insightful input, which has contributed to the improvement of our manuscript.

Location of changes in the revised manuscript (marked in blue): Please refer to the updated Table 1, updated Table 2, and the text on page 9, lines 471-480 for further clarifications.

Regarding the clinical application in patient triage, I wonder how the reading time was recorded

in radiologists since it is too long.

Response: Thank you for bringing this to our attention. In our study, we initially defined the 'reading time of radiologists' as the period from the beginning of the radiologist's image review until the issuance of the official report. This process entailed the review of CT scans, obtaining confirmation from at least two radiologists, and finally generating a report. We wholeheartedly agree with Reviewer 3's insightful suggestion that comparing the time taken by radiologists and LiAIDS to generate reports may not be equitable, as radiologists are expected to include extrahepatic findings in their reports. Consequently, we have chosen to remove any mention of the reading time of radiologists from our revised manuscript. Thank you for your insightful suggestion.

Please provide the criteria of 'correct detection' by LiAIDS: any criteria of dice score or others regarding the FLL segmentation?

Response: Thank you for bringing this to our attention. We appreciate your feedback and the opportunity to clarify the criteria for "correct detection" by LiAIDS. In our study, the determination of correct detection is based on the overlap of bounding boxes between the detected lesions and the ground-truth. Specifically, if the Intersection over Union (IOU) value is greater than 0.3 and/or the Intersection over Minimum (IOM) value is greater than 0.5, it indicates a hit and is considered a correct detection. These threshold values are used to ensure a reasonable level of overlap between the detected lesions and the ground truth. We hope this clarification provides a better understanding of the evaluation criteria used in our study. Thank you for your valuable input, which has contributed to improving the clarity and accuracy of our manuscript.

Location of changes in the revised manuscript (marked in blue): Please refer to the text on pages 7-8, lines 369-374 for additional clarifications.

How was the sample size determined in non-inferiority trial of 183 pathologically confirmed high risk patients (p. 14)?

Response: Thank you for raising this point. We appreciate your feedback. As specified in the revised manuscript, the estimation of the sample size for the trial was based on a non-inferiority margin (δ) of 0.1, a significance level (α) of 0.05, a power ($1-\beta$) of 0.8, and an anticipated efficacy accuracy of 0.92, leading us to anticipate a need for 173 participants. Therefore, our study is deemed sufficiently powered, given that the actual enrollment of 183 participants surpassed this calculated necessity.

Location of changes in the revised manuscript (marked in blue): Please refer to the text on page 9, lines 439-445 for additional clarifications.

Results

As I commented earlier, the reading time of radiologists is unusually long. Does this the 'review time' for this study? In other words, the time to record annotate all FLLs? If so, it is not a reading time to generate report. In addition, if it is a measurement of radiologists to review CT scans and then generate report, the comparison between the radiologists and LiAIDS is not fair because radiologists should mention extrahepatic findings.

***Response:** Thank you for bringing this to our attention. In our study, we initially defined the 'reading time of radiologists' as the period from the beginning of the radiologist's image review until the issuance of the official report. This process entailed the review of CT scans, obtaining confirmation from at least two radiologists, and finally generating a report. We wholeheartedly agree with Reviewer 3's insightful suggestion that comparing the time taken by radiologists and LiAIDS to generate reports may not be equitable, as radiologists are expected to include extrahepatic findings in their reports. Consequently, we have chosen to remove any mention of the reading time of radiologists from our revised manuscript. Thank you for your insightful suggestion.*

Also, per-lesion accuracy does not often reflect clinical relevance. Multiple hepatic cysts does not matter and detecting/reporting all hepatic cysts are not recommended since it is clinically irrelevant. In addition, patients with multiple liver metastases, all lesion detection may not be relevant either for the same reason.

***Response:** Thank you for your insightful suggestion. We have taken it into consideration and made updates to the revised manuscript to include results on patient-level analysis. We believe that this addition will provide a more comprehensive understanding of the study findings and enhance the overall quality of the manuscript. We appreciate your feedback and contribution to improving the clarity and completeness of our research.*

***Location of changes in the revised manuscript (marked in blue):** Please refer to the updated Table 2 and the text on page 9, lines 471-480 for additional clarifications.*

And please clarify the definition of junior and senior radiologists. Radiologists implies that they are board certified, and '5 to 10 years' experience of radiology' should mean that they have 5 to 10 years' experience in liver imaging (reading the films alone for clinical purpose, not research purpose) after having radiology board. Do authors mean this, and 'junior radiologist' does not indicate radiology resident or fellows? I request authors clarify this and whether they are body radiologist or general radiologist as well since your description in Discussion.

Response: *The medical hierarchy in China is structured into four distinct levels: physician (equivalent to resident), attending physician, associate chief physician, and chief physician. In this study, we adopted the duration of practice in abdominal imaging diagnosis as our primary classification criterion. Specifically, junior radiologists are general radiologists with 5-10 years of experience in abdominal imaging diagnosis, which includes physicians and attending physicians. Conversely, senior radiologists are general radiologists with 10-20 years of experience, and this group comprises associate chief physicians and chief physicians.*

Location of changes in the revised manuscript (marked in blue): *Please refer to the text on page 9, lines 463-470 for the specific clarifications.*

In addition, the results of non-inferiority study cannot be easily interpreted without number of each FLLs and the size of the tumor in addition to numbers of FLLs per patient.

Response: *Thank you for your valuable suggestion. We have taken your feedback into consideration and included additional information on the distribution of lesion size across all patients as well as the distribution of lesion number in each type across all patients in the non-inferiority study. These additions enhance the comprehensiveness of our study and provide a clearer understanding of the findings. We greatly appreciate your insightful input and your contribution to improving the quality of our manuscript.*

Location of changes in the revised manuscript (marked in blue): *Please refer to the newly added Figure 7 B and the text on page 9 lines 430-439 for additional clarifications.*

Discussion

Authors claimed that junior radiologists review the image initially and senior radiologists confirm it. I do not know much about Chinese radiology training system, but in most countries, this is only for radiology residents or fellows. No board-certified radiologists work in this manner. So it cannot be generalized and should be removed.

Response: *Thank you for your insightful suggestion. We have carefully considered your feedback and decided to remove the mentioned part from the revised manuscript. Your input is greatly appreciated, and we value your contribution to enhancing the quality of our work.*

Once again, we would like to express our sincere appreciation for the time and effort you dedicated to reviewing our manuscript. Your constructive comments and suggestions have been instrumental in improving the quality and clarity of our work. We have carefully incorporated your feedback into the revised version, and we hope that it now meets your expectations. We eagerly await your favorable response.

Thank you for your continued support.

Sincerely,

Xiujun Cai, PhD, Professor,
Department of General Surgery,
Sir Run Run Shaw Hospital,
Zhejiang University School of Medicine,
Hangzhou, China
Email: srrsh_cxj@zju.edu.cn

Reviewers' Comments:

Reviewer #1:

Remarks to the Author:

The authors have adequately addressed my critiques.

It's interesting to see that human+AI collaboration improved human performance for diagnosing liver lesions on CT. However, none of the radiologists aided by AI outperformed AI alone. This point needs to be discussed and how do you envision the AI model could be used and deployed in practice?

Other than that, I have no further comments.

Reviewer #2:

Remarks to the Author:

The authors have addressed my previous concerns. One minor comment is that Figure 3a-3e, and Figure 5f still has in the legends "Marco Average" instead of "Macro Average".

Reviewer #4:

Remarks to the Author:

Thank you for your extensive work. I am satisfied with the changes and responses provided.

Response Letter

12/3/2023

Dear Reviewer #1, Reviewer#2, and Reviewer #4,

We extend our heartfelt appreciation for your dedicated efforts and valuable feedback on our manuscript, "A Multicenter Study of a Clinically Applicable AI System for Automated Detection and Diagnosis of Focal Liver Lesions" (Manuscript ID: NCOMMS-23-00871A). Your insightful comments have been instrumental in enhancing the quality of our research.

Enclosed, please find the revised manuscript, featuring tracked changes for ease of reference. Below, we have provided detailed responses to each of your comments.

REVIEWERS' COMMENTS

Reviewer #1 (Remarks to the Author):

Comment: The authors have adequately addressed my critiques.

***Response:** We are grateful for your positive feedback and appreciate your constructive suggestions.*

Comment: It's interesting to see that human+AI collaboration improved human performance for diagnosing liver lesions on CT. However, none of the radiologists aided by AI outperformed AI alone. This point needs to be discussed and how do you envision the AI model could be used and deployed in practice?

***Response:** We appreciate your attention to the crucial aspect of AI and human collaboration in diagnostic processes. In the revised manuscript, we have elaborated on this topic, particularly focusing on the integration of LiAIDS with radiologist expertise (refer to Page 6, Lines 257-261 and 279-283, highlighted in blue). Our findings demonstrate that radiologists, irrespective of their experience levels, showed enhanced performance when assisted by LiAIDS. Specifically, junior radiologists achieved results comparable to LiAIDS alone, while senior radiologists' performance either matched or surpassed that of LiAIDS when using the tool.*

Additionally, we have addressed the practical deployment and application of the AI model in

clinical settings (see Page 107, Lines 509-516, highlighted in blue). This section details our vision for integrating LiAIDS into routine clinical practice, emphasizing its potential to augment the diagnostic accuracy and efficiency of radiologists.

Your feedback has been instrumental in enriching our manuscript, particularly in terms of discussing the real-world applicability and benefits of AI-assisted diagnostics. Thank you.

Comment: Other than that, I have no further comments.

Response: *We sincerely appreciate your positive and encouraging comments and endorsements expressed above.*

Reviewer #2 (Remarks to the Author):

Comment: The authors have addressed my previous concerns. One minor comment is that Figure 3a-3e, and Figure 5f still has in the legends "Marco Average" instead of "Macro Average".

Response: *We appreciate your attention to detail. The necessary corrections have been made in the aforementioned figures.*

Reviewer #4 (Remarks to the Author):

Comment: Thank you for your extensive work. I am satisfied with the changes and responses provided.

Response: *We are thankful for your positive endorsement of our revisions.*

In conclusion, we reiterate our gratitude for your thorough review and constructive comments. Your contributions have significantly improved both the quality and clarity of our manuscript.

Sincerely,

Xiujun Cai, PhD, Professor,
Department of General Surgery,
Sir Run Run Shaw Hospital,
Zhejiang University School of Medicine,
Hangzhou, China

Email: srrsh_cxj@zju.edu.cn